# Type-to-Track: Retrieve Any Object via Prompt-based Tracking

**Pha Nguyen[1], Kha Gia Quach[2], Kris Kitani[3], Khoa Luu[1]**

[1] CVIU Lab, University of Arkansas   [2] pdActive Inc.   [3] Robotics Institute, Carnegie Mellon University

[1]`{panguyen, khoaluu}@uark.edu`   [2]`kquach@ieee.org`   [3]`kkitani@cs.cmu.edu`

`uark-cviu.github.io/Type-to-Track`

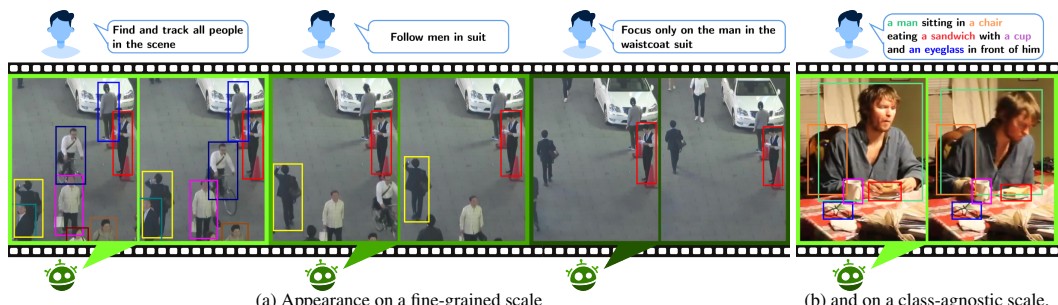

(a) Appearance on a fine-grained scale                   (b) and on a class-agnostic scale.

Figure 1: An example of the responsive *Type-to-Track*. The user provides a video sequence and a prompting request. During tracking, the system is able to discriminate appearance attributes to track the target subjects accordingly and iteratively responds to the user's tracking request. Each box color represents a unique identity.

## Abstract

One of the recent trends in vision problems is to use natural language captions to describe the objects of interest. This approach can overcome some limitations of traditional methods that rely on bounding boxes or category annotations. This paper introduces a novel paradigm for Multiple Object Tracking called *Type-to-Track*, which allows users to track objects in videos by typing natural language descriptions. We present a new dataset for that Grounded Multiple Object Tracking task, called *GroOT*, that contains videos with various types of objects and their corresponding textual captions describing their appearance and action in detail. Additionally, we introduce two new evaluation protocols and formulate evaluation metrics specifically for this task. We develop a new efficient method that models a transformer-based eMbed-ENcoDE-extRact framework (*MENDER*) using the third-order tensor decomposition. The experiments in five scenarios show that our *MENDER* approach outperforms another two-stage design in terms of accuracy and efficiency, up to 14.7% accuracy and $4\times$ speed faster.

## 1   Introduction

Tracking the movement of objects in videos is a challenging task that has received significant attention in recent years. Various methods have been proposed to tackle this problem, including deep learning techniques. However, despite these advances, there is still room for improvement in intuitiveness and responsiveness. One potential way to improve object tracking in videos is to incorporate user input into the tracking process. Traditional Visual Object Tracking (VOT) methods typically require

37th Conference on Neural Information Processing Systems (NeurIPS 2023).

Table 1: Comparison of current datasets. # denotes the number of the corresponding item. **Bold** numbers are the best number in each sub-block, while  highlighted  numbers are the best across all sub-blocks.

| Datasets | Task | NLP | #Videos | #Frames | #Tracks | #AnnBoxes | #Words | #Settings |
|---|---|---|---|---|---|---|---|---|
| **OTB100** [8] | SOT | ✗ | 100 | 59K | 100 | 59K | - | - |
| **VOT-2017** [9] | SOT | ✗ | 60 | 21K | 60 | 21K | - | - |
| **GOT-10k** [10] | SOT | ✗ | 10K | 1.5M | 10K | 1.5M | - | - |
| **TrackingNet** [11] | SOT | ✗ | **30K** | **14.43M** | 30K | **14.43M** | - | - |
| **MOT17** [12] | **MOT** | ✗ | 14 | 11.2K | 1.3K | 0.3M | - | - |
| **TAO** [13] | **MOT** | ✗ | 1.5K | **2.2M** | 8.1K | 0.17M | - | - |
| **MOT20** [14] | **MOT** | ✗ | 8 | 13.41K | 3.83K | 2.1M | - | - |
| **BDD100K** [15] | **MOT** | ✗ | **2K** | 318K | **130.6K** | **3.3M** | - | - |
| **LaSOT** [6] | SOT | ✓ | 1.4K | **3.52M** | 1.4K | **3.52M** | 9.8K | 1 |
| **TNL2K** [7] | SOT | ✓ | 2K | 1.24M | 2K | 1.24M | 10.8K | 1 |
| **Ref-DAVIS** [16] | VOS | ✓ | 150 | 94K | 400+ | - | 10.3K | **2** |
| **Refer-YTVOS** [17] | VOS | ✓ | **4K** | 1.24M | **7.4K** | 131K | **158K** | **2** |
| **Ref-KITTI** [18] | **MOT** | ✓ | 18 | 6.65K | - | - | 3.7K | 1 |
| **GroOT (Ours)** | **MOT** | ✓ | **1,515** | **2.25M** | **13.3K** | **2.57M** | **256K** | **5** |

users to manually select objects in the video by points [1], bounding boxes [2, 3], or trained object detectors [4, 5]. Thus, in this paper, we introduce a new paradigm, called *Type-to-Track*, to this task that combines responsive typing input to guide the tracking of objects in videos. It allows for more intuitive and conversational tracking, as users can simply type in the name or description of the object they wish to track, as illustrated in Fig. 1. Our intuitive and user-friendly *Type-to-Track* approach has numerous potential applications, such as surveillance and object retrieval in videos.

We present a new Grounded Multiple Object Tracking dataset named *GroOT* that is more advanced than existing tracking datasets [6, 7]. *GroOT* contains videos with various types of multiple objects and detailed textual descriptions. It is $2\times$ larger and more diverse than any existing datasets, and it can construct many different evaluation settings. In addition to three easy-to-construct experimental settings, we propose two new settings for prompt-based visual tracking. It brings the total number of settings to five, which will be presented in Section 5. These new experimental settings challenge existing designs and highlight the potential for further advancements in our proposed research topic.

In summary, this work addresses the use of natural language to guide and assist the Multiple Object Tracking (MOT) tasks with the following contributions. First, a novel paradigm named *Type-to-Track* is proposed, which involves responsive and conversational typing to track any objects in videos. Second, a new *GroOT* dataset is introduced. It contains videos with various types of objects and their corresponding textual descriptions of 256K words describing definition, appearance, and action. Next, two new evaluation protocols that are tracking by *retrieval prompts* and *caption prompts*, and three class-agnostic tracking metrics are formulated for this problem. Finally, a new transformer-based eMbed-ENcoDE-extRact framework (*MENDER*) is introduced with third-order tensor decomposition as the first efficient approach for this task. Our contributions in this paper include a novel paradigm, a rich semantic dataset, an efficient methodology, and challenging benchmarking protocols with new evaluation metrics. These contributions will be advantageous for the field of Grounded MOT by providing a valuable foundation for the development of future algorithms.

## 2 Related Work

### 2.1 Visual Object Tracking Datasets and Benchmarks

**Datasets.** To develop and train VOT models for the computer vision task of tracking objects in videos, various datasets have been created and widely used. Some of the most popular datasets for VOT are OTB [19, 8], VOT [9], GOT [10], MOT challenges [12, 14] and BDD100K [15]. Visual object tracking has two sub-tasks: *Single Object Tracking* (SOT) and *Multiple Object Tracking* (MOT). Table 1 shows that there is a wide variety of object tracking datasets in both types available, each with its own strengths and weaknesses. Existing datasets with NLP [6, 7] only support the SOT task, while our *GroOT* dataset supports MOT with approximately $2\times$ larger in description size.

**Benchmarks.** Current benchmarks for tracking can be broadly classified into two main categories: *Tracking by Bounding Box* and *Tracking by Natural Language*, depending on the type of initialization.

Table 2: Comparison of key features of tracking methods. **Cls-agn** is for class-agnostic, while **Feat** is for the approach of feature fusion and **Stages** indicates the number of stages in the model design incorporating NLP into the tracking task. **NLP** indicates how text is utilized for the tracker: *assist* (w/ box) or can *initialize* (w/o box).

| Approach | Task | NLP | Cls-agn | Feat | Stages |
|---|---|---|---|---|---|
| GTI [27] | SOT | assist | ✗ | concat | single |
| TransVLT [28] | SOT | assist | ✗ | attn | single |
| TrackFormer [4] | MOT | – | ✗ | – | – |
| MDETR+TFm | MOT | init | ✓ | attn | two |
| TransRMOT [18] | MOT | init | ✓ | attn | two |
| **MENDER** | **MOT** | init | ✓ | attn | single |

Table 3: Statistics of *GroOT*'s settings.

| Datasets | | #Videos | #Frames | #Tracks | #AnnBoxes | #Words | Parts |
|---|---|---|---|---|---|---|---|
| MOT17** | Train | 7 | 5,316 | 546* | 112,297* | 3,792 | (1) |
| | Test | 7 | 5,919 | 785* | 188,076* | 5,757 | (2) |
| | **Total** | 14 | 11,235 | 1,331* | 300,373* | 9,549 | |
| TAO** | Train | 500 | 764,526 | 2,645 | 54,639 | 19,222 | (3) |
| | Val | 993 | 1,460,666 | 5,485 | 113,112 | 39,149 | (4) |
| | Test | 914 | 2,221,846 | 7,972 | 164,650 | - | |
| | **Total** | 2,407 | 4,447,038 | 16,089 | 332,401 | 58,371 | |
| MOT20** | Train | 4 | 8,931 | 2,332* | 1,336,920* | - | (5) |
| | Test | 4 | 4,479 | 1,501* | 765,465* | - | (6) |
| | **Total** | 8 | 13,410 | 3,833* | 2,102,385* | - | |
| GroOT** | **nm** | 1,515 | 2,249,837 | 13,294 | 2,570,509 | 21,424 | *all* |
| | **syn** | 1,515 | 2,249,837 | 13,294 | 2,570,509 | 53,540 | *all* |
| | **def** | 1,515 | 2,249,837 | 13,294 | 2,570,509 | 99,218 | *all* |
| | **cap** | 1,507 | 2,236,427 | 9,461 | 468,124 | 67,920 | *w/o MOT20* |
| | **retr** | 993 | 1,460,666 | 1,952 | - | 13,935 | *uses (4)* |

*all* uses (1, 2, 3, 4, 5, 6) and *w/o MOT20* uses (1, 2, 3, 4).
* Statistics from the official site, including objects other than human.
** Creative Commons Attribution-NonCommercial-ShareAlike 3.0 License

Previous benchmarks [20, 19, 8, 9, 21, 22, 22, 23] were limited to test videos before the emergence of deep trackers. The first publicly available benchmarks for visual tracking were OTB-2013 [19] and OTB-2015 [8], consisting of 50 and 100 video sequences, respectively. GOT-10k [10] is a benchmark featuring 10K videos classified into 563 classes and 87 motions. TrackingNet [11], a subset of the object detection benchmark YT-BB [24], includes 31K sequences. Furthermore, there are long-term tracking benchmarks such as OxUvA [25] and LaSOT [6]. OxUvA spans 14 hours of video in 337 videos, comprising 366 object tracks. On the other hand, LaSOT [6] is a language-assisted dataset consisting of 1.4K sequences with 9.8K words in their captions. In addition to these benchmarks, TNL2K [7] includes 2K video sequences for natural language-based tracking and focuses on expressing the attributes. LaSOT [6] and TNL2K [7] support one benchmarking setting with their provided prompts, while our *GroOT* dataset supports five settings. Ref-KITTI [18] is built upon the KITTI [26] dataset and contains only two categories, including car and pedestrian, while our *GroOT* dataset focuses on category-agnostic tracking, and outnumbers the frames and settings.

A similar task with a different nomenclature to the Grounded MOT task is Referring Video Object Segmentation (Ref-VOS) [16, 17], which primarily measures the overlapping area between the ground truth and prediction for a single foreground object in each caption, with less emphasis on densely tracking multiple objects over time. In contrast, our proposed *Type-to-Track* paradigm is distinct in its focus on *responsively* and *conversationally* typing to track any objects in videos, requiring maintaining the temporal motions of multiple objects of interest.

## 2.2 Grounded Object Tracking

**Grounded Vision-Language Models** accurately map language concepts onto visual observations by understanding both vision content and natural language. For instance, visual grounding [29] seeks to identify the location of nouns or short phrases (such as a black hat or a blue bird) within an image. Grounded captioning [30, 31, 32] can generate text descriptions and align predicted words with object regions in an image. Visual dialog [33] enables meaningful dialogues with humans about visual content using natural, conversational language. Some visual dialog systems may incorporate referring expression recognition [34] to resolve expressions in questions or answers.

**Grounded Single Object Tracking** is limited to tracking a single object with box-initialized and language-assisted methods. The GTI [27] framework decomposes the tracking by language task into three sub-tasks: Grounding, Tracking, and Integration, and generates tubelet predictions frame-by-frame. AdaSwitcher [7] module identifies tracking failure and switches to visual grounding for better tracking. [35] introduce a unified system using attention memory and cross-attention modules with learnable semantic prototypes. Another transformer-based approach [28] is presented including a cross-modal fusion module, task-specific heads, and a proxy token-guided fusion module.

## 2.3 Discussion

Most existing datasets and benchmarks for object tracking are limited in their coverage and diversity of language and visual concepts. Additionally, the prompts in the existing Grounded SOT benchmarks do not contain variations in covering many objects in a single prompt, which limits the application of existing trackers in practical scenarios. To address this, we present a new dataset and benchmarking

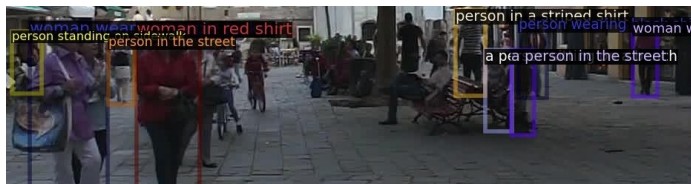

(a) Our MOT17 [12] subset sample with captions in both action and appearance types.

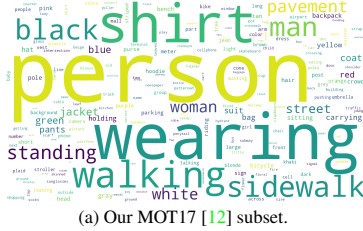

(a) Our MOT17 [12] subset.

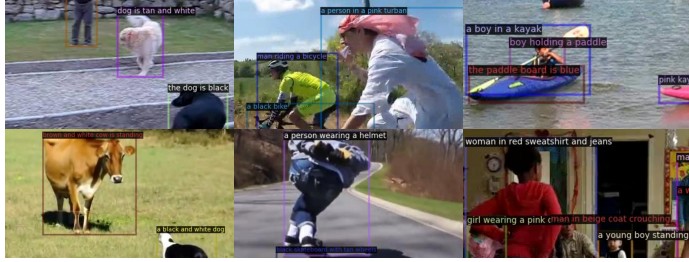

(b) Our TAO [13] subset samples with captions. **Best viewed in color and zoom in.**

Figure 2: Example sequences and annotations in our dataset.

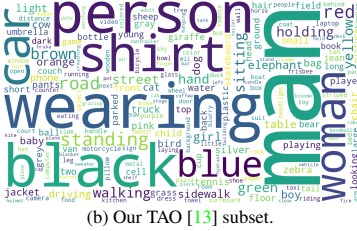

(b) Our TAO [13] subset.

Figure 3: Some words in our language description.

metrics to support the emerging trend of the Grounded MOT, where the goal is to align language descriptions with fine-grained regions or objects in videos.

As shown in Table 2, most of the recent methods for the Grounded SOT task are not class-agnostic, meaning they require prior knowledge of the object. GTI [27] and TransVLT [28] need to input the initial bounding box, while TrackFormer [4] need the pre-defined category. The operation used in [27] to fuse visual and textual features is *concatenation* which can only support prompts describing a single object. A Grounded MOT can be constructed by integrating a grounded object detector, i.e. MDETR [36], and an object tracker, i.e. TrackFormer [4]. However, this approach is low-efficient because the visual features have to be extracted multiple times. In contrast, our proposed MOT approach *MENDER* formulates third-order *attention* to adaptively focus on many targets, and it is an efficient *single-stage* and *class-agnostic* framework. The scope of *class-agnostic* in our approach is constructing a large vocabulary of concepts via a visual-textual corpus, following [37, 38, 39].

## 3 Dataset Overview

### 3.1 Data Collection and Annotation

Existing object tracking datasets are typically designed for specific types of video scenes [40, 41, 42, 43, 44, 2]. To cover a diverse range of scenes, *GroOT* was created using official videos and bounding box annotations from the MOT17 [12], TAO [13], and MOT20 [14]. The MOT17 dataset comprises 14 sequences with diverse environmental conditions such as crowded scenes, varying viewpoints, and camera motion. The TAO dataset is composed of videos from seven different datasets, such as the ArgoVerse [45] and BDD [15] datasets containing outdoor driving scenes, while LaSOT [6] and YFCC100M [46] datasets include in-the-wild internet videos. Additionally, the AVA [47], Charades [48], and HACS [49] datasets include videos depicting human-human and human-object interactions. By combining these datasets, *GroOT* covers multiple types of scenes and encompasses a wide range of 833 objects. This diversity allows for a wide range of object classes with captions to be included, making it an invaluable resource for training and evaluating visual grounding algorithms.

We release our textual description annotations in COCO format [50]. Specifically, a new key `'captions'` which is a list of strings is attached to each `'annotations'` item in the official annotation. In the MOT17 subset, we attempt to maintain two types of caption for well-visible objects: one describes the *appearance* and the other describes the *action*. For example, the caption for a well-visible person might be [`'a man wearing a gray shirt'`, `'person walking on the street'`] as shown in Fig. 2a. However, 10% of tracklets only have one caption type, and 3% do not have any captions due to their low visibility. The physical characteristics of a person or their personal accessories, such as their clothing, bag color, and hair color are considered to be part of their appearance. Therefore, the appearance captions include verbs `'carrying'` or `'holding'` to describe personal accessories. In the TAO subset, objects other than humans have one caption

describing appearance, for instance, ['a red and black scooter']. Objects that are human have the same two types of captions as the MOT17 subset. An example is shown in Fig. 2b. These captions are consistently annotated throughout the tracklets. Fig. 3 is the word-cloud visualization of our annotations.

## 3.2 *Type-to-Track* Benchmarking Protocols

Let $\mathbf{V}$ be a video sample lasts $t$ frames, where $\mathbf{V} = \left\{ \mathbf{I}_t \mid t < |\mathbf{V}| \right\}$ and $\mathbf{I}_t$ be the image sample at a particular time step $t$. We define a request prompt $\mathbf{P}$ that describes the objects of interest, and $\mathbf{T}_t$ is the set of tracklets of interest up to time step $t$. The *Type-to-Track* paradigm requires a tracker network $\mathcal{T}(\mathbf{I}_t, \mathbf{T}_{t-1}, \mathbf{P})$ that efficiently take into account $\mathbf{I}_t$, $\mathbf{T}_{t-1}$, and $\mathbf{P}$ to produce $\mathbf{T}_t = \mathcal{T}(\mathbf{I}_t, \mathbf{T}_{t-1}, \mathbf{P})$. To advance the task of multiple object retrieval, another benchmarking set is created in addition to the *GroOT* dataset. While training and testing sets follow a *One-to-One* scenario, where each caption describes a single tracklet, the new retrieval set contains prompts that follow a *One-to-Many* scenario, where a short prompt describes multiple objects. This scenario highlights the need for diverse methods to improve the task of multiple object retrieval. The retrieval set is provided with a subset of tracklets in the TAO validation set and three custom *retrieval prompts* that change throughout the tracking process in a video $\{\mathbf{P}_{t_1=0}, \mathbf{P}_{t_2}, \mathbf{P}_{t_3}\}$, as depicted in Fig. 1(a). The *retrieval prompts* are generated through a semi-automatic process that involves: (i) selecting the most commonly occurring category in the video, and (ii) cascadingly filtering to the object that appears for the longest duration. In contrast, the *caption prompts* are created by joining tracklet captions in the scene and keeping it consistent throughout the tracking period. We name these two evaluation scenarios as *tracklet captions* **cap** and *object retrieval* **retr** . With three more easy-to-construct scenarios, five scenarios in total will be studied for the experiments in Section 5. Table 3 presents the statistics of the five settings, and the data portions are highlighted in the corresponding colors.

## 3.3 Class-agnostic Evaluation Metrics

As indicated in [51], long-tailed classification is a very challenging task in imbalanced and large-scale datasets such as TAO. This is because it is difficult to distinguish between similar fine-grained classes, such as bus and van, due to the class hierarchy. Additionally, it is even more challenging to treat every class independently. The traditional method of evaluating tracking performance leads to inadequate benchmarking and undesired tracking results. In our *Type-to-Track* paradigm, the main task is not to classify objects to their correct categories but to retrieve and track the object of interest. Therefore, to alleviate the negative effect, we reformulate the original per-category metrics of MOTA [52], IDF1 [53], HOTA [54] into class-agnostic metrics:

$$\text{MOTA} = \frac{1}{|CLS^n|} \sum_{cls}^{CLS^n} \left( 1 - \frac{\sum_t (\text{FN}_t + \text{FP}_t + \text{IDS}_t)}{\sum_t \text{GT}_t} \right)_{cls}, \text{CA-MOTA} = 1 - \frac{\sum_t (\text{FN}_t + \text{FP}_t + \text{IDS}_t)_{CLS^1}}{\sum_t (\text{GT}_{CLS^1})_t} \tag{1}$$

$$\text{IDF1} = \frac{1}{|CLS^n|} \sum_{cls}^{CLS^n} \left( \frac{2 \times \text{IDTP}}{2 \times \text{IDTP} + \text{IDFP} + \text{IDFN}} \right)_{cls}, \quad \text{CA-IDF1} = \frac{(2 \times \text{IDTP})_{CLS^1}}{(2 \times \text{IDTP} + \text{IDFP} + \text{IDFN})_{CLS^1}} \tag{2}$$

$$\text{HOTA} = \frac{1}{|CLS^n|} \sum_{cls}^{CLS^n} \left( \sqrt{\text{DetA} \cdot \text{AssA}} \right)_{cls}, \quad \text{CA-HOTA} = \sqrt{(\text{DetA}_{CLS^1}) \cdot (\text{AssA}_{CLS^1})} \tag{3}$$

where $CLS^n$ is the category, set size $n$ is reduced to 1 by combining all elements: $CLS^n \rightarrow CLS^1$.

# 4 Methodology

## 4.1 Problem Formulation

Given the image $\mathbf{I}_t$ and the request prompt $\mathbf{P}$ describing the objects of interest, which can adaptively change between $\{\mathbf{P}_{t_1}, \mathbf{P}_{t_2}, \mathbf{P}_{t_3}\}$ in the **retr** setting, and $K$ is the prompt's length $|\mathbf{P}| = K$, let $enc(\cdot)$ and $emb(\cdot)$ be the visual encoder and the word embedding model to extract features of image tokens and prompt tokens, respectively. The resulting outputs, $enc(\mathbf{I}_t) \in \mathbb{R}^{M \times D}$ and

$emb(\mathbf{P}) \in \mathbb{R}^{K \times D}$, where $D$ is the length of feature dimensions. A list of region-prompt associations $\mathbf{C}_t$, which contains objects' bounding boxes and their confident scores, can be produced by Eqn. (4):

$$\mathbf{C}_t = \underset{\gamma}{dec}\Big(enc(\mathbf{I}_t)\,\bar{\times}\,emb(\mathbf{P})^{\intercal}, enc(\mathbf{I}_t)\Big) = \Big\{\mathbf{c}_i = (c_x, c_y, c_w, c_h, c_{conf})_i \mid i < M\Big\}_t \quad (4)$$

where $(\bar{\times})$ is an operation representing the region-prompt correlation, that will be elaborated in the next section, $\underset{\gamma}{dec}(\cdot, \cdot)$ is an object decoder taking the similarity and the image features to decode to object locations, thresholded by a scoring parameter $\gamma$ (i.e. $c_{conf} \geq \gamma$). For simplicity, the cardinality of the set of objects $|\mathbf{C}_t| = M$, implying each image token produces one region-text correlation.

We define $\mathbf{T}_t = \Big\{\mathbf{tr}_j = (tr_x, tr_y, tr_w, tr_h, tr_{conf}, tr_{id})_j \mid j < N\Big\}_t$ produced by the tracker $\mathcal{T}$, where $N = |\mathbf{T}_t|$ is the cardinality of current tracklets. $i$, $j$, $k$, and $t$ are consistently denoted as indexers for objects, tracklets, prompt tokens, and time steps for the rest of the paper.

**Remark 1** *Third-order Tensor Modeling. Since the Type-to-Track paradigm requires three input components $\mathbf{I}_t$, $\mathbf{T}_{t-1}$, and $\mathbf{P}$, an **auto-regressive single-stage end-to-end framework** can be formulated via third-order tensor modeling.*

To achieve this objective, a combination of initialization, object decoding, visual encoding, feature extraction, word embedding, and aggregation can be formulated as in Eqn. (5):

$$\mathbf{T}_t = \begin{cases} initialize(\mathbf{C}_t) & t = 0 \\ \underset{\gamma}{dec}\Big(\mathbf{1}_{D \times D \times D} \times_1 enc(\mathbf{I}_t) \times_2 ext(\mathbf{T}_{t-1}) \times_3 emb(\mathbf{P}), enc(\mathbf{I}_t)\Big) & \forall t > 0 \end{cases} \quad (5)$$

where $ext(\cdot)$ denotes the visual feature extractor of the set of tracklets, $ext(\mathbf{T}_{t-1}) \in \mathbb{R}^{N \times D}$, $\mathbf{1}_{D \times D \times D}$ is an all-ones tensor has size $D \times D \times D$, ( $\times_n$ ) is the $n$-mode product of the third-order tensor [55] to aggregate many types of token[1], and $initialize(\cdot)$ is the function to ascendingly assign unique identities to tracklets for the first time those tracklets appear.

Let $T \in \mathbb{R}^{M \times N \times K}$ be the resulting tensor $T = \mathbf{1}_{D \times D \times D} \times_1 enc(\mathbf{I}_t) \times_2 ext(\mathbf{T}_{t-1}) \times_3 emb(\mathbf{P})$. The objective function can be expressed as the log softmax of the positive region-tracklet-prompt triplet over all possible triplets, defined in Eqn. (6):

$$\theta^*_{enc,ext,emb} = \arg\max_{\theta_{enc,ext,emb}} \left(\log\Big(\frac{\exp(T_{ijk})}{\sum_l^K \sum_n^N \sum_m^M \exp(T_{lnm})}\Big)\right) \quad (6)$$

where $\theta$ denotes the network's parameters, the combination of the $i^{th}$ image token, the $j^{th}$ tracklet, and the $k^{th}$ prompt token is the correlated triplet.

In the next subsection, we elaborate our model design for the tracking function $\mathcal{T}(\mathbf{I}_t, \mathbf{T}_{t-1}, \mathbf{P})$, named *MENDER*, as defined in Eqn. (5), and loss functions for the problem objective in Eqn. (6).

## 4.2 MENDER for Multiple Object Tracking by Prompts

The correlation in Eqn. (5) has the cubic time and space complexity $\mathcal{O}(n^3)$, which can be intractable as the input length grows and hinder the model scalability.

**Remark 2** *Correlation Simplification. Since both $enc(\cdot)$ and $ext(\cdot)$ are visual encoders, the region-prompt correlation can be equivalent to the tracklet-prompt correlation. Therefore, the region-tracklet-prompt correlation tensor $T$ can be simplified to lower the computation footprint.*

To design that goal, the extractor and encoder share network weights for computational efficiency:

$$ext(\mathbf{T}_{t-1})_j = ext\Big(\{\mathbf{tr}_j\}_{t-1}\Big) = \Big\{enc(\mathbf{I}_{t-1})_i \colon \mathbf{c}_i \mapsto \mathbf{tr}_j\Big\}, \text{ therefore } \Big((T_{:j:})_{t-1} = (T_{i::})_t\Big) \colon \mathbf{c}_i \mapsto \mathbf{tr}_j{}^{[2]} \quad (7)$$

where $T_{:j:}$ and $T_{i::}$ are lateral and horizontal slices. In layman's terms, the **region-prompt** correlation at the time step $t-1$ is equivalent to the **tracklet-prompt** correlation at the time step $t$, as visualized in Fig. 4(a). Therefore, one practically needs to model the **region-tracklet** and **tracklet-prompt**

---

[1] implemented by a single Python code with Numpy: `np.einsum('ai, bj, ck -> abc', P, I, T)`.

[2] If $\mathbf{P}$ changes, the equivalence still holds true, see Appendix for the full algorithm.

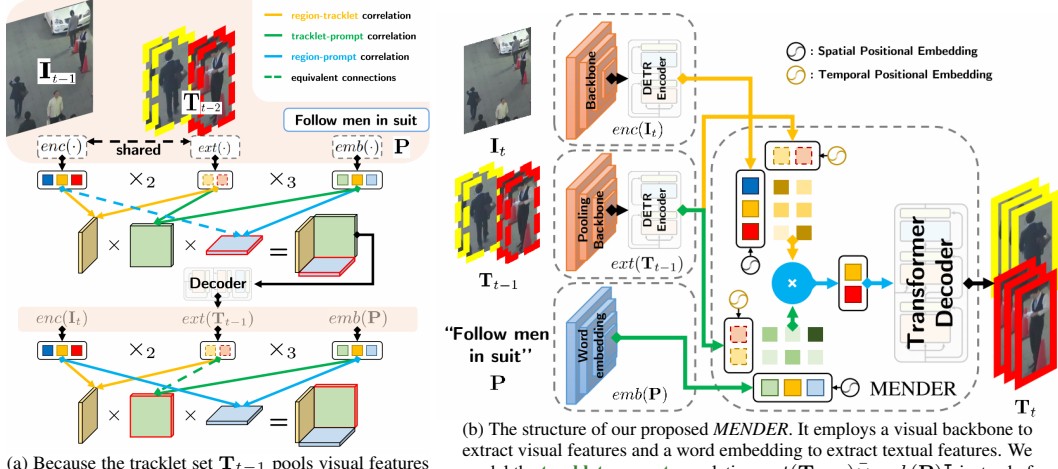

(a) Because the tracklet set $\mathbf{T}_{t-1}$ pools visual features of the image $\mathbf{I}_{t-1}$, the **region-prompt** is equivalent with **tracklet-prompt** (only need to filter unassigned objects).

(b) The structure of our proposed *MENDER*. It employs a visual backbone to extract visual features and a word embedding to extract textual features. We model the **tracklet-prompt** correlation $ext(\mathbf{T}_{t-1}) \bar{\times} emb(\mathbf{P})^{\mathsf{T}}$ instead of the **region-prompt** to avoid unnecessary computation caused by no-object tokens [56]. **Best viewed in color and zoom in.**

Figure 4: The *auto-regressive* manner takes advantage of the equivalent components. Simplifying the correlation in (a) turns the solution to *MENDER* in (b), and reduces complexity to $\mathcal{O}(n^2)$ where $n$ denotes the size of tokens.

correlations which reduces time and space complexity from $\mathcal{O}(n^3)$ to $\mathcal{O}(n^2)$, significantly lowering computation footprint. We alternatively rewrite the decoding step in Eqn. (5) as follows:

$$\mathbf{T}_t = \underset{\gamma}{dec}\left( \left( enc(\mathbf{I}_t) \bar{\times} ext(\mathbf{T}_{t-1})^{\mathsf{T}} \right) \times \left( ext(\mathbf{T}_{t-1}) \bar{\times} emb(\mathbf{P})^{\mathsf{T}} \right), enc(\mathbf{I}_t) \right) \quad \forall t > 0 \tag{8}$$

**Correlation Representations.** In our approach, the correlation operation $(\bar{\times})$ is modelled by the *multi-head cross-attention* mechanism [57], as depicted in Fig. 4(b). The attention matrix can be computed as:

$$\sigma(\mathbf{X}) \bar{\times} \sigma(\mathbf{Y}) = \mathcal{A}_{\mathbf{X}|\mathbf{Y}} = \text{softmax}\left( \frac{\left( \sigma(\mathbf{X}) \times W_Q^{\mathbf{X}} \right) \times \left( \sigma(\mathbf{Y}) \times W_K^{\mathbf{Y}} \right)^{\mathsf{T}}}{\sqrt{D}} \right) \tag{9}$$

where $\mathbf{X}$ and $\mathbf{Y}$ tokens are one of these types: region, tracklet, prompt. $\sigma(\cdot)$ is one of the operations $enc(\cdot)$, $emb(\cdot)$, $ext(\cdot)$ as the corresponding operation to $\mathbf{X}$ or $\mathbf{Y}$. Superscript $W_Q$, $W_K$, and $W_V$ are the projection matrices corresponding to $\mathbf{X}$ or $\mathbf{Y}$ as in the attention mechanism.

Then, the attention weight from the image $\mathbf{I}_t$ to the prompt $\mathbf{P}$ are computed by the matrix multiplication for $\mathcal{A}_{\mathbf{I}|\mathbf{T}}$ and $\mathcal{A}_{\mathbf{T}|\mathbf{P}}$ to aggregate the information from two matrices as in Eqn. (8). The result is the matrix $\mathcal{A}_{\mathbf{I}|\mathbf{T} \times \mathbf{T}|\mathbf{P}} = \mathcal{A}_{\mathbf{I}|\mathbf{T}} \times \mathcal{A}_{\mathbf{T}|\mathbf{P}}$ that shows the correlation between each input or output. Then, the resulting attention matrix $\mathcal{A}_{\mathbf{I}|\mathbf{T} \times \mathbf{T}|\mathbf{P}}$ is used to produce the object representations at time $t$:

$$\mathbf{Z}_t = \mathcal{A}_{\mathbf{I}|\mathbf{T} \times \mathbf{T}|\mathbf{P}} \times \left( emb(\mathbf{P}) \times W_V^{\mathbf{P}} \right) + \mathcal{A}_{\mathbf{I}|\mathbf{T}} \times \left( ext(\mathbf{T}_{t-1}) \times W_V^{\mathbf{T}} \right) \tag{10}$$

**Object Decoder** $dec(\cdot)$ utilizes context-aware features $\mathbf{Z}_t$ that are capable of preserving identity information while adapting to changes in position. The tracklet set $\mathbf{T}_t$ is defined in the *auto-regressive* manner to adjust to the movements of the object being tracked as in Eqn. (8). For decoding the final output at any frame, the decoder transforms the object representation by a 3-layer FFN to predict bounding boxes and confidence scores for frame $t$:

$$\mathbf{T}_t = \left\{ \mathbf{tr}_j = (tr_x, tr_y, tr_w, tr_h, tr_{conf})_j \right\}_t \overset{tr_{conf} \geq \gamma}{=} \text{FFN}\left( \mathbf{Z}_t + enc(\mathbf{I}_t) \right) \tag{11}$$

where the identification information of tracklets, represented by $tr_{id}$, is not determined directly by the FFN model. Instead, the $tr_{id}$ value is set when the tracklet is first initialized and maintained till its end, similar to *tracking-by-attention* approaches [4, 58, 59, 60].

### 4.3 Training Losses

To achieve the training objective function as in Eqn. (6), we formulate the objective function into two loss functions $L_{\mathbf{I}|\mathbf{T}}$ and $L_{\mathbf{T}|\mathbf{P}}$ for correlation training and one loss $L_{GIoU}$ for decoder training:

$$\mathcal{L} = \gamma_{\mathbf{T}|\mathbf{P}} L_{\mathbf{T}|\mathbf{P}} + \gamma_{\mathbf{I}|\mathbf{T}} L_{\mathbf{I}|\mathbf{T}} + \gamma_{GIoU} L_{GIoU} \qquad (12)$$

where $\gamma_{\mathbf{T}|\mathbf{P}}$, $\gamma_{\mathbf{I}|\mathbf{T}}$, and $\gamma_{GIoU}$ are corresponding coefficients, which are set to $0.3$ by default.

**Alignment Loss** $L_{\mathbf{T}|\mathbf{P}}$ is a contrastive loss, which is used to assure the alignment of the ground-truth object feature and caption pairs $(\mathbf{T}, \mathbf{P})$ which can be obtained in our dataset. There are two alignment losses used, one for all objects normalized by the number of positive prompt tokens and the other for all prompt tokens normalized by the number of positive objects. The total loss can be expressed as:

$$L_{\mathbf{T}|\mathbf{P}} = $$
$$-\frac{1}{|\mathbf{P}^+|} \sum_k^{|\mathbf{P}^+|} \log\left( \frac{\exp\left( ext(\mathbf{T})_j^\intercal \times emb(\mathbf{P})_k \right)}{\sum_l^K \exp\left( ext(\mathbf{T})_j^\intercal \times emb(\mathbf{P})_l \right)} \right) - \frac{1}{|\mathbf{T}^+|} \sum_j^{|\mathbf{T}^+|} \log\left( \frac{\exp\left( emb(\mathbf{P})_k^\intercal \times ext(\mathbf{T})_j \right)}{\sum_l^N \exp\left( emb(\mathbf{P})_k^\intercal \times ext(\mathbf{T})_l \right)} \right)$$
$$(13)$$

where $\mathbf{P}^+$ and $\mathbf{I}^+$ are the sets of positive prompts and image tokens corresponding to the selected $enc(\mathbf{I})_i$ and $emb(\mathbf{P})_k$, respectively.

**Objectness Losses.** To model the track's temporal changes, our network learns from training samples that capture both appearance and motion generated by two adjacent frames:

$$L_{\mathbf{I}|\mathbf{T}} = -\sum_j^N \log\left( \frac{\exp\left( ext(\mathbf{T})_j^\intercal \times enc(\mathbf{I})_i \right)}{\sum_l^N \exp\left( ext(\mathbf{T})_j^\intercal \times enc(\mathbf{I})_l \right)} \right) \quad \text{, and} \quad L_{GIoU} = \sum_j^N \ell_{GIoU}(\mathbf{tr}_j, \mathbf{obj}_i) \qquad (14)$$

$L_{\mathbf{I}|\mathbf{T}}$ is the log-softmax loss to guide the tokens' alignment as similar to Eqn. (13). In the $L_{GIoU}$ loss, $\mathbf{obj}_i$ is the ground truth object corresponding to $\mathbf{tr}_j$. The optimal assignment between $\mathbf{tr}_j$ or $\mathbf{obj}_i$ to the ground truth object is computed efficiently by the Hungarian algorithm, following DETR [56]. $\ell_{GIoU}$ is the Generalized IoU loss [61].

## 5 Experimental Results

### 5.1 Implementation Details

**Experimental Scenarios.** We create three types of prompt: *category name* **nm** , *category synonyms* **syn** , *category definition* **def** . One *tracklet captions* **cap** scenario is constructed by our detailed annotations and one more *objects retrieval* **retr** scenario is given in our custom request prompts as described in Subsec. 3.2. The dataset contains 833 classes, each has a name and a corresponding set of synonyms that are different names for the same category, such as [man, woman, human, pedestrian, boy, girl, child] for person. Additionally, each category is described by a *category definition* sentence. This definition makes the model deal with the variations in the text prompts. We join the names, synonyms, definitions, or captions and filter duplicates to construct the prompt. Trained models use as the same type as testing. We annotated the raw tracking data of the best-performant tracker (i.e., BoT-SORT [62] at 80.5% MOTA and 80.2% IDF1) at the time we constructed experiments and used it as the sub-optimal ground truth of MOT17 and MOT20 (parts *(2, 4)* in Table 3). That is also the raw data we used to evaluate all our ablation studies.

**Datasets and Metrics.** RefCOCO+ [63] and Flickr30k [64] serve as pre-trained datasets for acquiring a vocabulary of visual-textual concepts [37]. The $ext(\cdot)$ operation is not involved in this training step. After obtaining a pre-trained model from RefCOCO+ and Flickr30k, we train and evaluate our model for the proposed *Type-to-Track* task on all five scenarios on our *GroOT* dataset and the first-three scenarios for MOT20 [14]. The tracking performance is reported in class-agnostic metrics CA-MOTA, CA-IDF1, and CA-HOTA as in Subsec. 3.3 and mAP50 as defined in [13].

**Tokens Production.** $emb(\cdot)$ utilizes RoBERTa [65] to convert the text input into a sequence of numerical tokens. The tokens are fed into the RoBERTa-base model for text encoding using a

Table 4: Ablation studies. **sim** indicates whether the correlation is the *simplified* Eqn. (8) or the Eqn. (5). See 5.1 for the abbreviations. The two first settings get only one word for the request prompt, therefore, tensor $T$ is an unsqueezed matrix, resulting in no difference in **nm** (✗) vs (✓), and **syn** (✗) vs (✓).

| P | sim | CA-MOTA | CA-IDF1 | MT | IDs | mAP | FPS |
|---|---|---|---|---|---|---|---|
| | | GroOT - MOT17 Subset | | | | | |
| nm | ✗/✓ | 67.00 | 71.20 | 544 | 1352 | 0.876 | 10.3 |
| syn | ✗/✓ | 65.10 | 71.10 | 554 | 1348 | 0.874 | 10.3 |
| def | ✗ | 67.00 | 72.10 | 556 | 1343 | 0.876 | 5.8 |
| | ✓ | **67.30** | **72.40** | **568** | **1322** | **0.877** | **10.3** |
| cap | ✗ | 58.20 | 53.20 | **289** | 1751 | 0.674 | 3.4 |
| | ✓ | **59.50** | **54.80** | 201 | **1734** | **0.688** | **7.8** |
| | | GroOT - TAO Subset | | | | | |
| nm | ✓ | 27.30 | 37.20 | 3523 | 4284 | 0.212 | 11.2 |
| syn | ✓ | 25.70 | 36.10 | 3212 | 5048 | 0.198 | 11.2 |
| def | ✗ | 15.20 | 27.30 | 2452 | 6253 | 0.154 | 6.2 |
| | ✓ | **16.80** | **27.70** | **2547** | **6118** | **0.158** | **10.5** |
| cap | ✗ | 20.30 | 31.80 | 2943 | 5242 | **0.188** | 4.3 |
| | ✓ | **20.70** | **32.00** | **3103** | **5192** | 0.184 | **8.7** |
| retr | ✗ | 32.40 | 38.40 | 630 | 3238 | 0.423 | 7.6 |
| | ✓ | **32.90** | **39.30** | **645** | **3194** | **0.430** | **11.5** |
| | | GroOT - MOT20 Subset | | | | | |
| nm | ✗/✓ | 72.40 | 67.50 | 823 | 2498 | 0.826 | 7.6 |
| syn | ✗/✓ | 70.90 | 65.30 | 809 | 2509 | 0.823 | 7.6 |
| def | ✗ | 72.90 | 67.70 | **823** | 2489 | **0.826** | 4.3 |
| | ✓ | **72.10** | **67.10** | 812 | **2503** | 0.825 | **7.6** |

Table 5: Comparisons to the two-stage baseline design. In each dataset, the from-top-to-bottom scenarios are **syn**, **def**, **cap** and **retr**. **Best viewed in color.**

| Approach | CA-MOTA | CA-IDF1 | MT | IDs | mAP | FPS |
|---|---|---|---|---|---|---|
| | GroOT - MOT17 Subset | | | | | |
| MDETR + TFm | 62.60 | 64.70 | 519 | 1382 | 0.793 | 2.2 |
| **MENDER** | **65.10** | **71.10** | **554** | **1348** | **0.874** | **10.3** |
| MDETR + TFm | 62.60 | 64.70 | 519 | 1382 | 0.793 | 2.2 |
| **MENDER** | **67.30** | **72.40** | **568** | **1322** | **0.877** | **10.3** |
| MDETR + TFm | 44.80 | 45.20 | 193 | 1945 | 0.619 | 2.1 |
| **MENDER** | **59.50** | **54.80** | **201** | **1734** | **0.688** | **7.8** |
| | GroOT - TAO Subset | | | | | |
| MDETR + TFm | 21.30 | 33.20 | 2945 | 5834 | 0.184 | 3.1 |
| **MENDER** | **25.70** | **36.10** | **3212** | **5048** | **0.198** | **11.2** |
| MDETR + TFm | 14.60 | 21.40 | 1944 | 6493 | 0.137 | 3.1 |
| **MENDER** | **16.80** | **27.70** | **2547** | **6118** | **0.158** | **10.5** |
| MDETR + TFm | 15.30 | 23.60 | 2132 | 6354 | 0.156 | 3.0 |
| **MENDER** | **20.70** | **32.00** | **3103** | **5192** | **0.182** | **8.7** |
| MDETR + TFm | 25.70 | 26.40 | 513 | 3993 | 0.387 | 3.1 |
| **MENDER** | **32.90** | **39.30** | **645** | **3194** | **0.430** | **11.5** |
| | GroOT - MOT20 Subset | | | | | |
| MDETR + TFm | 61.20 | 60.40 | 784 | 2824 | 0.732 | 1.9 |
| **MENDER** | **70.90** | **65.30** | **809** | **2509** | **0.823** | **7.6** |
| MDETR + TFm | 68.00 | 66.30 | 763 | 2975 | 0.783 | 1.9 |
| **MENDER** | **72.10** | **67.10** | **812** | **2503** | **0.825** | **7.6** |

12-layer transformer network with 768 hidden units and 12 self-attention heads per layer. $enc(\cdot)$ is implemented using a ResNet-101 [66] as the backbone to extract visual features from the input image. The output of the ResNet is processed by a Deformable DETR encoder [67] to generate visual tokens. For each dimension, we use sine and cosine functions with different frequencies as positional encodings, similar to [68]. A feature resizer combining a list of (Linear, LayerNorm, Dropout) is used to map to size $D = 512$ for all token producers.

## 5.2 Ablation Study

**Comparisons in Different Scenarios.** Table 4 shows comparisons in the performance of different prompt inputs. For MOT17 and MOT20, the *category name* is 'person', while *category definition* is 'a human being'. Since the prompt by *category definition* is short, it does not differ much from the **nm** setting. However, the **syn** setting shuffles between some words, resulting in a slight decrease in CA-MOTA and CA-IDF1. The **cap** setting results in prompts that contain more diverse and complex vocabulary, and more context-specific information. It is more difficult for the model to accurately localize the objects and identify their identity within the image, as it needs to take into account a wider range of linguistic cues, resulting in a decrease in performance compared to **def** (59.5% CA-MOTA and 54.8% CA-IDF1 vs 67.3% CA-MOTA and 72.4% CA-IDF1 on MOT17).

For TAO, the **def** setting has a significant number of variations and many tenuous connections in the scene context, for example, 'an aircraft that has a fixed wing and is powered by propellers or jets' for the airplane category. Therefore, it results in a decrease in performance (16.8% CA-MOTA and 27.7% CA-IDF1) compared to **cap** (20.7% CA-MOTA and 32.0% CA-IDF1), because the **cap** setting is more specific on the object level than category level. The best performant setting is **nm** (27.3% CA-MOTA and 37.2% CA-IDF1), where names are combined.

**Simplied Attention Representations.** Table 4 also presents the effectiveness of different attention representations of the full tensor $T$ (denoted by ✗) and the simplified correlation (denoted by ✓). The performance is reported with frame per second (FPS), which is self-measured on one GPU NVIDIA RTX 3060 12GB. Overall, the performance of simplified correlation is witnessed with a superior speed of up to 2× (7.8 FPS vs 3.4 FPS of **cap** on MOT17 and 11.5 FPS vs 7.6 FPS of **retr** on TAO), resulting in and a slight increase in accuracy due to attention stability and precision gain.

Table 6: Comparisons to the state-of-the-art approaches on the *category name* **nm** setting.

| Approach | Cls-agn | CA-IDF1 | CA-MOTA | CA-HOTA | MT | ML | AssA | DetA | LocA | IDs |
|----------|---------|---------|---------|---------|-----|-----|------|------|------|-----|
| ByteTrack [69] | ✗ | 77.3 | 80.3 | 63.1 | 957 | 516 | 52.7 | 55.6 | 81.8 | 3,378 |
| TrackFormer [4] | ✗ | 68.0 | 74.1 | 57.3 | 1,113 | 246 | 54.1 | 60.9 | 82.8 | 2,829 |
| QuasiDense [70] | ✗ | 66.3 | 68.7 | 53.9 | 957 | 516 | 52.7 | 55.6 | 81.8 | 3,378 |
| CenterTrack [71] | ✗ | 64.7 | 67.8 | 52.2 | 816 | 579 | 51.0 | 53.8 | 81.5 | 3,039 |
| TraDeS [72] | ✗ | 63.9 | 69.1 | 52.7 | 858 | 507 | 50.8 | 55.2 | 81.8 | 3,555 |
| CTracker [73] | ✗ | 57.4 | 66.6 | 49.0 | 759 | 570 | 45.2 | 53.6 | 81.3 | 5,529 |
| **MENDER** | ✓ | 67.1 | 65.0 | 53.9 | 678 | 648 | 54.4 | 53.6 | 83.4 | 3,266 |

## 5.3 Comparisons with A Baseline Design

Due to the new proposed topic, no current work has the same scope or directly solves our problem. Therefore, we compare our proposed *MENDER* against a two-stage baseline tracker in Table 5. We use current SOTA methods to develop this approach, i.e., MDETR [36] for the grounded detector, while TrackFormer [4] for the object tracker. It is worth noting that our *MENDER* relies on direct regression to locate and track the object of interest, without the need for an explicit grounded object detection stage. Table 5 shows our proposed *MENDER* outperforms the baseline on both CA-MOTA and CA-IDF1 metrics in all four settings *category synonyms*, *category definition*, *tracklet captions* and *object retrieval* (25.7% vs. 21.3%, 16.8% vs. 14.6%, 20.7% vs. 15.3% and 32.9% vs. 25.7% CA-MOTA on TAO), while can maintain up to $4\times$ run-time speed (10.3 FPS vs 2.2 FPS). The results indicate that training a single-stage network enhances efficiency and reduces errors by avoiding separate feature extractions for both detection and tracking steps.

## 5.4 Comparisons with State-of-the-Art Approaches

The *category name* **nm** setting is also the official MOT benchmark. Table 6 is the comparison of our result on the *category name* setting on the official leaderboard of MOT17, compared with other state-of-the-art approaches, including ByteTrack [69] and TrackFormer [4]. Note that our proposed *MENDER* is one of the first attempts at the Grounded MOT task, not to achieve the top rankings on the general MOT leaderboard. In contrast, other SOTA approaches benefit from the efficient single-category design in their separate object detectors, while our single-stage design is agnostic to the category and for flexible textual input. Compared to TrackFormer [4], our proposed *MENDER* only demonstrates a marginal decrease in identity assignment (67.1% vs 68.0% CA-IDF1). The decrease in the CA-MOTA stems from our detector's design which integrates flexible input.

# 6 Conclusion

We have presented a novel problem of *Type-to-Track*, which aims to track objects using natural language descriptions instead of bounding boxes or categories, and a large-scale dataset to advance this task. Our proposed *MENDER* model reduces the computational complexity of third-order correlations by designing an efficient attention method that scales quadratically w.r.t the input sizes. Our experiments on three datasets and five scenarios demonstrate that our model achieves state-of-the-art accuracy and speed for class-agnostic tracking.

**Limitations.** While our proposed metrics effectively evaluate the proposed *Type-to-Track* problem, they may not be ideal for measuring precision-recall characteristics in retrieval tasks. Additionally, the lack of the question-answering task in data and problem formulation may limit the algorithm to not being able to provide language feedback such as clarification or alternative suggestions. Additional benchmarks incorporating question-answering are excellent research avenues for future work. While the performance of our proposed *MENDER* may not be optimal for well-defined categories, it paves the way for exploring new avenues in open vocabulary and open-world scenarios [74].

**Broader Impacts.** The *Type-to-Track* problem and the proposed *MENDER* model have the potential to impact various fields, such as surveillance and robotics, where recognizing object interactions is a crucial task. By reformulating the problem with text support, the proposed methodology can improve the intuitiveness and responsiveness of tracking, making it more practical for video input support in large-language models [75] and real-world applications similar to ChatGPT. However, it could bring potential negative impacts related to human rights by providing a video retrieval system via text.

**Acknowledgment.** This work is partly supported by NSF Data Science, Data Analytics that are Robust and Trusted (DART), and Google Initiated Research Grant. We also thank Utsav Prabhu and Chi-Nhan Duong for their invaluable discussions and suggestions and acknowledge the Arkansas High-Performance Computing Center for providing GPUs.

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
