# Type-to-Track: Retrieve Any Object via Prompt-based Tracking
## Supplementary

# Appendix

## 1    Dataset Taxonomy

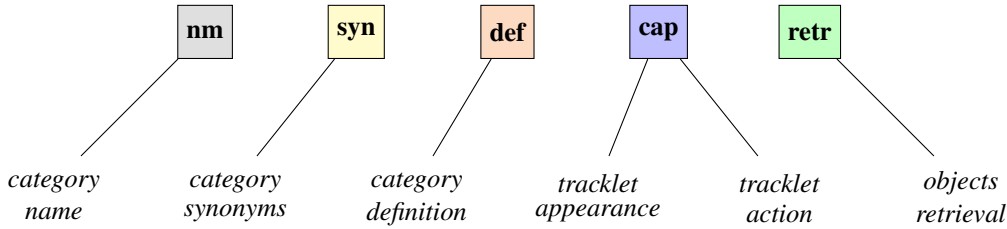

Figure 1: Full list of the types of caption to construct the request prompt for the corresponding settings.

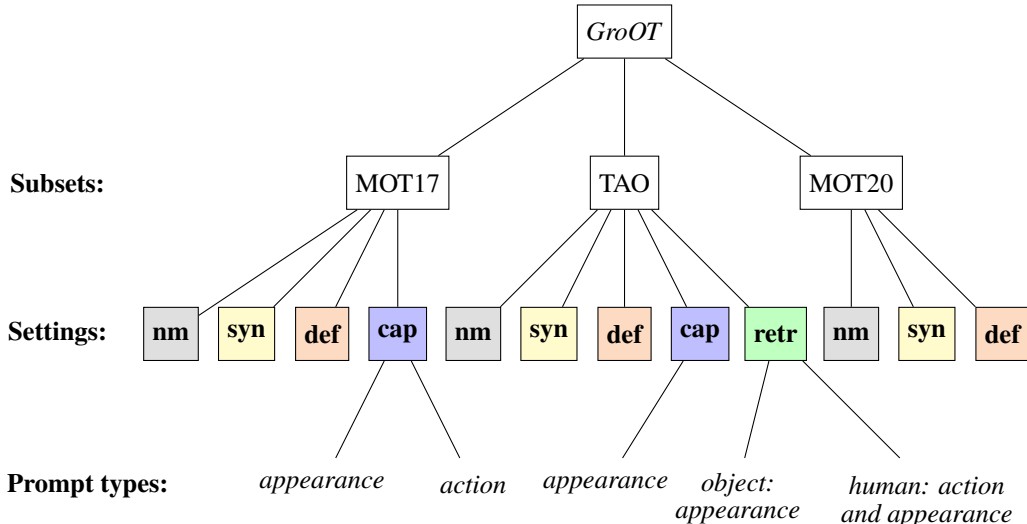

Figure 2: Types of prompt for the construction of our settings for each dataset.

We introduce two new evaluation scenarios **cap** and **retr** so that they are more specific on the object level than on the category level. It is because defining objects by category synonyms and category names and definition is insufficient to describe them accurately, leading to ambiguous results. The benchmarking sets can provide more accurate and meaningful evaluations of multiple object retrieval methods by focusing on the object level.

We include a comprehensive taxonomy of prompt types used to construct our settings. However, the **retr** setting on the MOT17 could not be constructed because test annotations for this dataset are unavailable. To construct this setting, bounding boxes will be filtered to the corresponding *retrieval prompt* when it changes. Section 2 describes how to construct this *retrieval prompt*. The MOT20 dataset requires extensive annotations and has many low-visible people due to the crowd view. Therefore, its annotations are not ready to be released at the moment.

## 2    Annotation Process

Instead of collecting new videos, we add annotations to the widely used MOT17 [1] and TAO [2] evaluation sets. These sets contain diverse and relatively long videos with fast-moving objects, camera motion, various object sizes, frequent object occlusions, scale changes, motion blur, and similar objects. Another advantage is that multiple objects are typically present throughout the entire sequence, which is desirable for long-term tracking scenarios.

We entrust ten professional annotators to annotate all frames. They use an interactive open-sourced annotation tool [3] that incorporates the K-nearest neighbor to speed up the annotation process of similar tracklets. All annotations are manually verified.

Then, we post-process the annotations to construct the *retrieval prompts* . Retrieval prompts are short phrases or sentences that retrieve relevant information from the video. The process of generating these prompts involves two main steps:

1. Select the most commonly occurring category in the video. It is done to ensure that the generated prompts are relevant to the video's content and capture the main objects or scenes in the video. For example, if the video is about a soccer game, the most commonly occurring category might be 'soccer players' or 'soccer ball'.

2. Filter the category selected in the first step to the object that appears for the longest duration. It is likely done to ensure the generated prompts are specific and focused on a particular object or scene in the video. For example, if the most commonly occurring category in a soccer game video is 'soccer players', the longest appearing player is selected as the focus of the retrieval prompt.

## 3    Data Format

### 3.1   categories

```
1  categories [{
2      'frequency': str,
3      'id': int,
4      'synset': str,
5      'image_count': int,
6      'instance_count': int,
7      'name':  str
8      'synonyms':  [str] ,
9      'def':  str
10  }]
```

The categories field of the annotation structure stores a mapping of category id to the category name, synonyms, and definitions. The categories field is structured as an array of dictionaries. Each dictionary in the array represents a single category.

The keys and values of the dictionary are:

- 'frequency': A string value that indicates the frequency of the category in the dataset.
- 'id': An integer value that represents the unique ID assigned to the category.
- 'synset': A string value that contains a unique identifier for the category.
- 'image_count': An integer value that indicates the number of images in the dataset that belong to the category.
- 'instance_count': An integer value that indicates the number of instances of the category that appear in the dataset.
- 'name': A string value that represents the name of the category.
- 'synonyms': An array of string values that contains synonyms of the category name.
- 'def': A string value that provides a definition of the category.

## 3.2 `annotations`

```
1  annotations[{
2      'id': int,
3      'image_id': int,
4      'category_id': int,
5      'scale_category': str,
6      'track_id': int,
7      'video_id': int,
8      'segmentation': [polygon],
9      'area': float,
10     'bbox': [x, y, width, height],
11     'iscrowd': 0 or 1,
12     'captions':  [str]
13 }]
```

An object instance annotation is a record that describes a single instance of an object in an image or video. It is structured as a dictionary containing a series of key-value pairs, where each key corresponds to a specific field in the annotation. The fields included in the annotation are:

- 'id': An integer value that represents the unique ID assigned to the annotation.
- 'image_id': An integer value that represents the ID of the image that the object instance is part of.
- 'category_id': An integer value that represents the ID of the category to which the object instance belongs.
- 'scale_category': A string value that represents the scale of the object instance with respect to the category.
- 'track_id': An integer value that represents the ID of the track to which the object instance belongs.
- 'video_id': An integer value that represents the ID of the video that the object instance is part of.
- 'segmentation': An array of polygon coordinates that represent the segmentation mask of the object instance.
- 'area': A float value that represents the area of the object instance.
- 'bbox': An array of four values that represent the bounding box coordinates of the object instance.
- 'iscrowd': A binary value (0 or 1) that indicates whether the object instance is a single object or a group of objects.
- 'captions': An array of string values that contains annotated textual descriptions of the object instance. The first caption is implicitly annotated as appearance, while the next one is action.

## 3.3 `images`

```
1  images[{
2      'id': int,
3      'frame_index': int,
4      'video_id': int,
5      'file_name': str,
6      'width': int,
7      'height': int,
8      'video': str,
9      'prompt':  str
10 }]
```

The `images` annotations are used to construct request prompts by using the image index at a particular timestamp. To do this, we use the 'images" field in the annotation structure, which contains information about the images in the dataset.

Each image in the dataset is represented as a dictionary object with the following fields:

- 'id': an integer ID for the image
- 'frame_index': an integer value representing the frame index or time stamp index of the image
- 'video_id': an integer ID for the video the image belongs to
- 'file_name': a string value representing the name of the image file
- 'width': an integer value representing the width of the image in pixels
- 'height': an integer value representing the height of the image in pixels
- 'video': a string value representing the name of the video the image belongs to
- 'prompt': a string value representing the request prompt for the video at a particular time stamp which is indexed by 'frame_index'.

The 'prompt' field is the key field used to construct the request prompt, and it is generated based on the information in the annotations for the objects in the image. Using the annotations to generate the prompt, it becomes possible to retrieve specific data about the objects in the image, such as their category, location, and size.

## 4 Examples

### 4.1 Data Samples

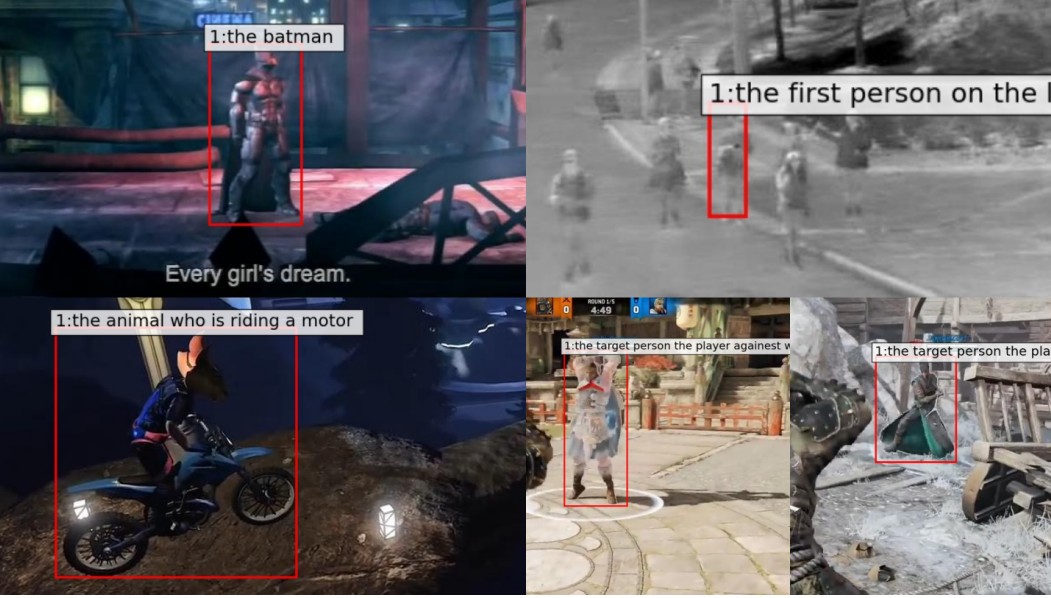

Figure 3: Samples in TNL2K [4] dataset. The annotations are not meaningful and not discriminative. This dataset also overlooks many moving objects that are present in the video but are not annotated.

In Fig. 3, we present some samples from the TNL2K [4] dataset. This dataset only contains SOT annotations, which are less meaningful than our dataset. For example, the annotations for some objects in the images, such as 'the batman', 'the first person on the left side', and 'the animal riding a motor', can be confusing for both viewers and algorithms. In some cases, the same caption describes two different objects. For instance, in a video game scene, two opponents are annotated with the same caption 'the target person the player against with'. Additionally, this dataset overlooks some large moving objects present in the video. Therefore,

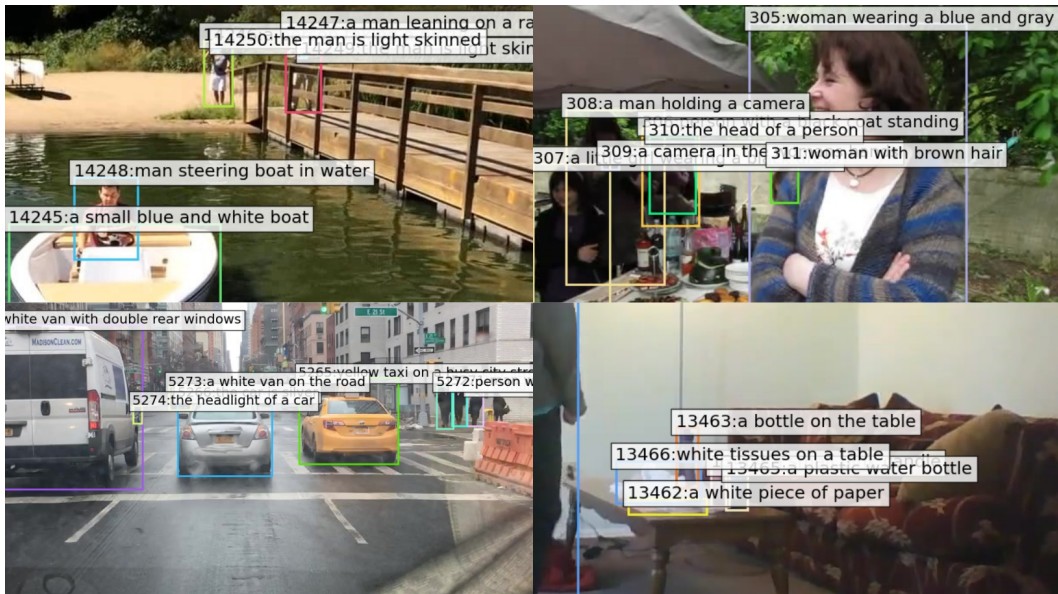

Figure 4: Samples in our *GroOT* dataset cover almost all moving objects with discriminative captions and a variety of object types. Labels are shown in the following format: `track_id:np.random.choice(captions)`.

while the TNL2K dataset provides some valuable data, it also has significant limitations regarding the clarity, discrimination, consistency of the annotations, and scope of the annotated objects.

On the other hand, Fig. 4 shows some samples from our *GroOT* dataset, which covers almost all moving objects in the video and provides distinct captions. The dataset includes a variety of object types and provides accurate and comprehensive annotations such as '`white tissues on a table`', '`a bottle on the table`', etc. It allows for more effective training and evaluation of Grounded MOT algorithms.

## 4.2 Annotations

Table 1: Examples of annotations in the *GroOT* dataset.

| MOT17 | |
|---|---|
| '`name`' | '`person`' |
| '`synonyms`' | [‘`baby`’, ‘`child`’, ‘`boy`’, ‘`girl`’, ‘`man`’, ‘`woman`’, ‘`perdestrian`’, ‘`human`’] |
| '`definition`' | '`a human being`' |
| '`captions`' | [‘`man walking on sidewalk`’, ‘`man wearing a orange shirt`’] |
| TAO | |
| '`name`' | '`backpack`' |
| '`synonyms`' | [‘`backpack`’, ‘`knapsack`’, ‘`packsack`’, ‘`rucksack`’, ‘`haversack`’] |
| '`definition`' | '`a bag carried by a strap on your back or shoulder`' |
| '`captions`' | [‘`a black colored bag`’, ‘`the bag is yellow in color`’] |

Table 1 provides examples of annotations in the *GroOT* dataset. For instance, the MOT17 subset has annotations for the object class *'person'* with synonyms including '`baby`', '`child`', '`boy`', '`girl`', '`man`', '`woman`', '`pedestrian`' and '`human`'. The definition for this class is '`a human being`' and example captions could include '`man walking on sidewalk`' or '`man wearing an orange shirt`'. On the other hand, the TAO subset has annotations for the object class '`backpack`', with synonyms such as '`knapsack`', '`packsack`', '`rucksack`', and '`haversack`'. The definition for this class is '`a bag carried by a strap on your back or shoulder`' and example captions could include '`a black colored bag`' or '`the bag is yellow in color`'.

## 4.3 Run-time Prompts

Table 2: Examples of constructing request prompts in the proposed evaluation settings.

| | MOT17 | MOT20 |
|---|---|---|
| **nm** | 'person' | 'person' |
| **syn** | ['man', 'woman'] | ['man', 'woman'] |
| **def** | ['a human being'] | ['a human being'] |
| **cap** | ['a man in a suit', 'man wearing an orange shirt', 'a woman in a black shirt and pink skirt'] | N/A |

| | TAO |
|---|---|
| | *Example 1* |
| **nm** | ['bus', 'bicycle', 'person'] |
| **syn** | ['autobus', 'bicycle', 'perdestrian'] |
| **def** | ['a vehicle carrying many passengers; used for public transport', 'a motor vehicle with two wheels and a strong frame', 'a human being'] |
| **cap** | ['a black van', 'silver framed bicycle', 'person wearing black pants'] |
| **retr** | 'people crossing the street' |
| | *Example 2* |
| **nm** | ['man', 'cup', 'chair', 'sandwich', 'eyeglass'] |
| **syn** | ['person', 'cup', 'chair', 'sandwich', 'spectacles'] |
| **def** | ['a human being', 'a small open container usually used for drinking; usually has a handle', 'a seat for one person, with a support for the back', 'two (or more) slices of bread with a filling between them', 'optical instrument consisting of a frame that holds a pair of lenses for correcting defective vision'] |
| **cap** | ['a man wearing a gray shirt', 'a white cup on the table', 'wooden chair in white room', 'the sandwich is triangle', 'an eyeglasses on the table'] |
| **retr** | 'a man sitting on a chair eating a sandwich with a cup and an eyeglass in front of him' |

Table 2 presents examples of how the annotations described earlier can be used to construct request prompts during runtime. In MOT17 and MOT20 subsets, the only category is 'person' with randomly selected synonyms 'man' and 'woman' and the definition 'a human being'. The captions for the MOT17 subset include 'a man in a suit', 'man wearing an orange shirt' and 'a woman in a black shirt and pink skirt', while the captions for the MOT20 subset are not annotated.

For TAO subset, the categories in the first example on a driving scene include 'bus', 'bicycle' and 'person' with the synonyms being 'autobus', 'bicycle' and 'pedestrian', respectively. The definitions for these categories are 'a vehicle carrying many passengers; used for public transport', 'a motor vehicle with two wheels and a strong frame' and 'a human being', respectively. The captions include 'a black van', 'silver framed bicycle', and 'person wearing black pants', while the retrieval is 'people crossing the street'.

Example 2 shows another example of how annotations can be used to construct request prompts. The categories in this example include 'man', 'cup', 'chair', 'sandwich' and 'eyeglass' with the synonyms being 'person', 'cup', 'chair', 'sandwich' and 'spectacles', respectively. The definitions for these categories are 'a human being', 'a small open container usually used for drinking; usually has a handle', 'a seat for one person,

with a support for the back', 'two (or more) slices of bread with a filling
between them' and 'optical instrument consisting of a frame that holds a pair
of lenses for correcting defective vision', respectively. The joint captions include
'a man wearing a gray shirt', 'a white cup on the table', 'wooden chair in
white room', 'the sandwich is triangle' and 'an eyeglass on the table', while the
retrieval prompt is 'a man sitting on a chair eating a sandwich with a cup and an
eyeglass in front of him'.

## 5    Methodolody

### 5.1    3D Transformers

**Third-order Tensor Modeling.** Our design of third-order tensor to handle three input components
$\mathbf{I}_t$, $\mathbf{T}_{t-1}$, and $\mathbf{P}$ influences the design of a novel 3D Transformer. Current temporal visual-textual
modeling [5, 6, 7] uses two dimensions and computes interactions between video and text features
spanned over the temporal domain. However, our approach is different because it handles three
components individually, which allows for more flexibility and a more nuanced understanding of
the data. By modeling as the $n$-mode product of the third-order tensor to aggregate many types of
tokens, we have presented a general methodology that can be scaled to multi-modality. Using the 3D
Transformer model, which allows for interactions between these features over time, can improve the
performance of multi-modal models by enabling them to consider a wider range of input features and
their temporal dependencies. Therefore, our design of third-order tensor modeling has the potential
for further research in multi-modality applications.

### 5.2    Symmetric Alignment Loss

Both the *Alignment Loss* $L_{\mathbf{T}|\mathbf{P}}$ and the *Objectness Loss* $L_{\mathbf{I}|\mathbf{T}}$ are log-softmax loss functions because
they both aim to maximize the similarity of the alignments. The *Alignment Loss* has two terms, one
for all objects normalized by the number of positive prompt tokens and the other for all prompt tokens
normalized by the number of positive objects. This way, the loss is symmetric and penalizes both
misalignments equally, especially for *different modalities*.

On the other hand, the *Objectness Loss* only computes from one side and is not necessarily symmetric
because there is a *single modality* in this case. It only needs to focus on the quality of the object
alignment to the image and does not need to consider the quality of the image alignment to the object.
Consider two objects, $A$ and $B$ are equivalent. If we want to maximize the similarity between object
$A$ and the correct alignment, we can compute the loss on $A$ with $B$ or $B$ with $A$. The similarity
between object $A$ and object $B$ is maximized in both cases.

### 5.3    Algorithmic Complexity

This section briefly discusses the computational complexity of our approach. The attention part
in recent MOT approaches [8, 9] has the quadratic time and space complexity $\mathcal{O}(M \times N)$[1], or
similarly [10] has $\mathcal{O}((M + N)^2)$. In our *Grounded* setting, by incorporating the textual request, the
complexity even scales up to $\mathcal{O}(M \times N \times K)$ for the $n$-mode product of the third-order tensor in
Eqn. (8).

We assume the network structure is fixed; hence the dimensions of the embedding feature vectors and
layers are constant numbers. Therefore, the complexity of one network pass is constant, i.e., $\mathcal{O}(1)$.

It can be observed that the overall complexity of our model depends on the *MENDER* combining
**region-tracklet** and **tracklet-prompt** correlations. From Eqn. (7), the time complexity depends
on the matrix multiplication operation. In contrast, $\mathcal{A}_{\mathbf{I}|\mathbf{T} \times \mathbf{T}|\mathbf{P}} = \mathcal{A}_{\mathbf{I}|\mathbf{T}} \times \mathcal{A}_{\mathbf{T}|\mathbf{P}}$ is performed on
scalars, not performed on token vectors, so it is ignored from the calculation. Mathematically, the
time complexity of our *MENDER* will be equivalent to $\mathcal{O}(M \times N + N \times K)$ and will quadratically
grow for the size of tokens. Simplifying the third-order correlation turns the solution to *MENDER*
and reduces the complexity from $\mathcal{O}(M \times N \times K)$ to $\mathcal{O}(M \times N + N \times K)$.

---

[1] All notations are defined as the same as in the main paper.

Notice that our approach strengthens our method by using the same attention mechanism for many steps, including **region-tracklet** and **tracklet-prompt** correlations, updating tracklets and preserving tracklet identities.

## 6 Additional Details

### 6.1 Implementation Details

---

**Algorithm 1** The inference pipeline of *MENDER*

---

**Input:** Video $\mathbf{V}$, set of tracklets $\mathbf{T} \leftarrow \varnothing$, set of prompts $\{\mathbf{P}_{t_1=0}, \mathbf{P}_{t_2}, \mathbf{P}_{t_3}\}$, $\gamma = 0.7$,
$\gamma_{reassign} = 0.75$, $t_{tlr} = 30$

1: **for** $t \in \{0, \cdots, |\mathbf{V}| - 1\}$ **do**
2:    **if** $t \in \{t_1, t_2, t_3\}$ **then**
3:       Select $\mathbf{P} \leftarrow \mathbf{P}_t$
4:    **end if**
5:    Draw $\mathbf{I}_t \in \mathbf{V}$
6:    **if** $\mathbf{T} = \varnothing$ **then**
7:       **if** $t = 0$ **then**
8:          $\mathbf{T}_{inactive} \leftarrow \varnothing$
9:       **else**
10:          *% This case happens when $\mathbf{P}$ changed to a completely new prompt without covering any old tracklets, returning an empty $\mathbf{T}$ at a timestamp $t \geq 0$ in line 23. Then the reinitialization is performed as in line 13 to line 14.*
11:          Pass
12:       **end if**
13:       $\mathbf{C} \leftarrow \underset{\gamma}{dec}\Big(enc(\mathbf{I}_t)\bar{\times}emb(\mathbf{P})^\intercal, enc(\mathbf{I}_t)\Big)$
14:       $\mathbf{T} \leftarrow initialize(\mathbf{C}_t)$ *% Obtaining tracklet $tr_{id}$'s*
15:    **else**
16:       $\mathbf{T}_{prev} \leftarrow \mathbf{T} + \mathbf{T}_{inactive}$
17:       *% If $\mathbf{P}$ does not change or it covers a subset of the previous objects, our MENDER forward has the ability to attend to the correct targets.*
18:       $\mathbf{T} \leftarrow \underset{\gamma}{dec}\Bigg(\Big(enc(\mathbf{I}_t)\bar{\times}ext(\mathbf{T}_{prev})^\intercal\Big) \times \Big(ext(\mathbf{T}_{prev})\bar{\times}emb(\mathbf{P})^\intercal\Big), enc(\mathbf{I}_t)\Bigg)$
19:       *% Obtaining tracklet $tr_{id}$'s*
20:       `matched_pairs, unmatched_lists` $\leftarrow cascade\_matching(\mathbf{T}, \mathbf{T}_{prev}, \gamma_{reassign})$
21:       `m_new, m_old` $\leftarrow$ `matched_pairs`
22:       `unm_new, unm_old` $\leftarrow$ `unmatched_lists`
23:       $\mathbf{T} \leftarrow update(\mathbf{T}[\texttt{m\_new}], \mathbf{T}_{prev}[\texttt{m\_old}]) + initialize\Big(\mathbf{T}[\texttt{unm\_new}]\Big)$
24:       $\mathbf{T}_{inactive} \leftarrow remove\_deprecation(\mathbf{T}_{inactive}, t_{tlr}) + \mathbf{T}_{prev}[\texttt{unm\_old}]$
25:    **end if**
26: **end for**

---

**Pseudo-Algorithm.** Alg. 1 is the pseudo-code for our *MENDER* algorithmic design, a Grounded Multiple Object Tracker that performs online multiple object tracking via text initialization. The pseudo-code provides a high-level overview of the steps involved in our *MENDER* method.

**Prompt Change without Losing Track.** If $\mathbf{P}$ changes to a new prompt between $\{\mathbf{P}_{t_1}, \mathbf{P}_{t_2}, \mathbf{P}_{t_3}\}$ that still covers a subset of the objects from the previous prompt, then the **region-prompt** correlation is still partially equivalent to the **tracklet-prompt** correlation. In this case, our *MENDER* can still attend to the correct targets even with the new prompt because it is trained to maximize the correct pairs which are influenced by the *Alignment Loss* and *Objectness Loss*.

However, if the prompt $\mathbf{P}$ changes entirely and no longer covers any of the objects from the previous prompt, then our *MENDER* needs to reinitialize the process by recomputing the **region-prompt**. It means that the algorithm needs to start over with the new **region-prompt** correlation and determine which objects to attend to, as in line 13 to line 14.

Table 3: Traditional metrics struggle to evaluate tracking performance due to uneven datasets and misclassified categories, leading to biased and inferior results.

| P | sim. | MOTA | IDF1 | CA-MOTA | CA-IDF1 | MT | IDs | mAP | FPS |
|---|---|---|---|---|---|---|---|---|---|
| **GroOT** - MOT17 Subset | | | | | | | | | |
| nm | ✗/✓ | 67.00 | 71.20 | 67.00 | 71.20 | 1344 | 1352 | 0.876 | 10.3 |
| syn | ✗/✓ | 65.10 | 71.10 | 65.10 | 71.10 | 1354 | 1348 | 0.874 | 10.3 |
| def | ✗ | 67.00 | 72.10 | 67.00 | 72.10 | 1356 | 1343 | 0.876 | 5.8 |
| | ✓ | **67.30** | **72.40** | **67.30** | **72.40** | **1368** | **1322** | **0.877** | **10.3** |
| cap | ✗ | 58.20 | 53.20 | 58.20 | 53.20 | 889 | 1751 | 0.674 | 3.4 |
| | ✓ | **59.50** | **54.80** | **59.50** | **54.80** | **801** | **1734** | **0.688** | **7.8** |
| **GroOT** - TAO Subset | | | | | | | | | |
| nm | ✓ | 3.10 | -53.20 | 27.30 | 37.20 | 4523 | 4284 | 0.212 | 11.2 |
| syn | ✓ | 3.00 | -57.10 | 25.70 | 36.10 | 4212 | 5048 | 0.198 | 11.2 |
| def | ✗ | 1.70 | -62.10 | 15.20 | 27.30 | 3452 | 6253 | 0.154 | 6.2 |
| | ✓ | 1.70 | -62.10 | **16.80** | **27.70** | **3547** | **6118** | **0.158** | **10.5** |
| cap | ✗ | 1.90 | -62.00 | 20.30 | 31.80 | 3943 | 5242 | 0.188 | 4.3 |
| | ✓ | 1.90 | -60.20 | **20.70** | **32.00** | **4103** | **5192** | **0.184** | **8.7** |
| retr | ✗ | 4.50 | -45.60 | 32.40 | 38.40 | 630 | 3238 | 0.423 | 7.6 |
| | ✓ | 4.50 | -45.60 | **32.90** | **39.30** | **645** | **3194** | **0.430** | **11.5** |
| **GroOT** - MOT20 Subset | | | | | | | | | |
| nm | ✗/✓ | 72.40 | 67.50 | 72.40 | 67.50 | 823 | 2498 | 0.826 | 7.6 |
| syn | ✗/✓ | 70.90 | 65.30 | 70.90 | 65.30 | 809 | 2509 | 0.823 | 7.6 |
| def | ✗ | 72.90 | 67.70 | 72.90 | 67.70 | 823 | 2489 | 0.826 | 4.3 |
| | ✓ | **72.10** | **67.10** | **72.10** | **67.10** | **812** | **2503** | **0.825** | **7.6** |

**Tracklets Management.** Our approach involves the *tracking-by-attention* paradigm [10, 9] that enables us to re-identify tracklets for a short period without requiring any specific re-identification training. It can be achieved by decoding tracklet features for a maximum number of $t_{tlr}$ tolerant frames. These tracklets are considered inactive during this tolerance, but they can still contribute to output trajectories when their re-assignment score exceeds $\gamma_{reassign}$.

**Training Process.** We follow the same training setting as [11] with a batch size of 4, 40 epochs, and different learning rates for the word embedding model, and the rest of the network, specifically, the learning rates are 0.00005 and 0.0001, respectively. We configure different max numbers for each token type: 250 for text queries, 500 for image queries, and 500 for tracklet queries. The training takes four days for MOT17 and seven days for MOT20 and TAO on 4 GPUs NVIDIA A100.

**Text Tokenizer.** *MENDER* employs RoBERTa Tokenizer [12] to convert textual input into a sequence of text tokens. It is done by dividing the text into a sequence of subword units using a pre-existing vocabulary. Each subword is then mapped to a unique numerical token ID using a lookup table. The tokenizer adds special tokens [CLS] and [SEP] to the beginning and end of the sequence, respectively. To encode the prompt for **def** and **cap** settings, the [CLS] token is used to represent each sentence in the prompt list, as in Table 1 and Table 2. For **nm** and **syn**, we join the words by '. ' and use the word features, following [13].

## 6.2 Negative Effects of the Long-tail Challenge on Tracking

The imbalance in the TAO's distribution negatively affects the performance of tracking algorithms and the evaluation of tracking metrics. Here are the negative effects of the long-tail problem on large-scale tracking datasets:

**Inaccurate Classification.** Large-scale tracking datasets like TAO contain numerous rare and semantically similar categories [14]. The classification performance for these categories is inaccurate due to the challenges of imbalanced datasets and distinguishing fine-grained classes [15, 16]. The inaccurate classification results in suboptimal tracking, where objects may be misclassified. It hinders the accurate evaluation of tracking algorithms, as classification is a prerequisite for conducting association and evaluating tracking performance.

**Suboptimal Tracking.** Current MOT methods and metrics typically associate objects with the same class predictions. In the case of large-scale datasets with inaccurate classification, this association strategy leads to suboptimal tracking. Even if the tracker localizes and tracks the object perfectly, it still receives a low score if the class prediction is wrong. As a result, the performance of trackers in tracking rare or semantically similar classes becomes negligible, and the performance of dominant classes dominates the evaluation.

**Inadequate Benchmarking.** The prevalent strategies in MOT evaluation group tracking results based on class labels and evaluate each class separately. However, this approach leads to inadequate benchmarking in large-scale datasets with inaccurate classification. Trackers performing well in localization and association but with inaccurate class predictions may receive low scores, even though their tracking results are valuable. For example, the trajectories of wrongly classified or unknown objects can still be helpful for tasks such as collision avoidance in autonomous vehicles [15].

Table 3 presents our findings which indicate that the performance of the Grounded MOT system is very poor on the traditional benchmarking metrics (0.17% to 0.45% MOTA and -45.60% to -62.10% IDF1 on TAO). The benchmarking metrics for this task should be designed to differentiate between the two tasks of classification and tracking. By separating these tasks, the CA-MOTA and CA-IDF1 can help to provide a more accurate assessment of tracking performance.

# 7 Qualitative Results

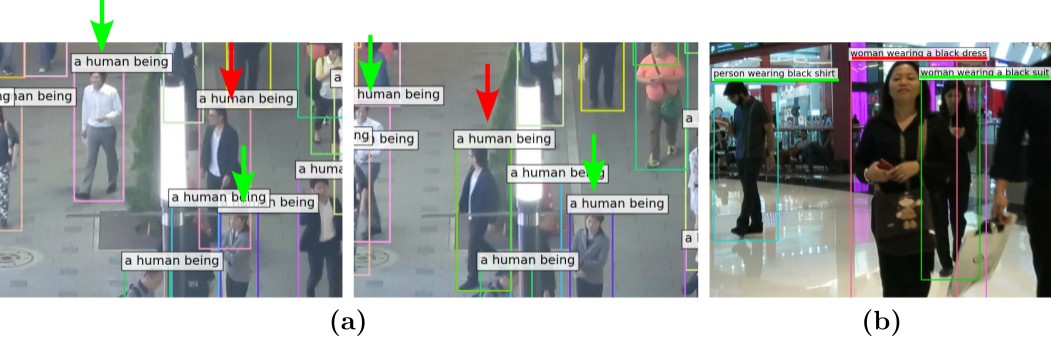

(a)                                                                                                     (b)

Figure 5: Qualitative results using detailed prompts. Each box color represents a unique tracklet identity. (a) **Green arrows** indicate true positive tracklets, while **red arrows** indicate false negative tracklets. (b) **Green lines** indicate correct attended caption of each tracklet, while the **red line** indicate the incorrect attended caption.

Fig. 5 shows two qualitative results in the Grounded Multiple Object Tracking problem with detailed request prompts. Fig. 5(a) is the **def** setting and Fig. 5(b) is the **cap** setting. See the supplementary video for more qualitative results.

**Failed Cases.** Fig. 5 also shows some failed cases of our *MENDER*. Fig. 5(a) indicates IDSwitch error by the **red arrows**. We also map the result tracklets to their attended caption. Fig. 5(b) shows the incorrect attended caption, which is highlighted by the **red line**.