# OpenReview forum: "Type-to-Track: Retrieve Any Object via Prompt-based Tracking"
_NeurIPS.cc/2023/Conference — NeurIPS 2023 poster_

### Official Review · Reviewer_j89C · 2023-07-03

**Soundness:** 3 good
**Presentation:** 3 good
**Contribution:** 3 good
**Rating:** 5
**Confidence:** 5

**Summary:**

This paper provides a new paradigm for MOT, using natural language descriptions to track objects of interest in videos. To accomplish it, the authors collected and annotated various object and scene descriptions. Meanwhile, the new evaluation protocols are also formulated following the evaluation of the original MOT. The authors also provide a new method, showing good performance in experiments.

**Strengths:**

* This work has an intuitive and good motivation to solve MOT using natural language.
* This paper is well organized, which contains collecting new benchmarks, designing effective methods, and conducting experiments.
* More importantly, the authors provide many visualization results, which are interesting.


**Weaknesses:**

* First of all, the main idea of ‘Type-to-Track’ has a similar idea to Referring Multi-Object Tracking [1] (released at Arxiv2023.03, accepted at CVPR2023), which both use language description to prompt multiple objects. It is important to discuss and compare the related work.
* I’m curious about the annotation details of the descriptions. Are they manually annotated? Or they are annotated by the current techniques of GPT?
* In Table 3, does the number of words represent the number of descriptions? Or the words are words of descriptions?
* The evaluation of caption prompts confuses me. The crucial words of created captions have one-to-one object matching with the grounded objects. So each object is evaluated individually?
* This paper is not easy to follow and sometimes overclaimed on some details, e.g., third-order tensor modeling (using three inputs), and correlation simplification (using separate attention). Besides, the authors define many new concepts but lack details and explanations, e.g., the single-stage end-to-end framework. How to deal with new entrance or exit objects?
* How about the visualization performance of the proposed method?

[1] Referring Multi-Object Tracking, CVPR2023.


**Questions:**

See the weakness section.

**Limitations:**

The authors have already provided the limitations in the paper.

---

> ### Author Rebuttal · Authors · 2023-08-05
>
> We express our gratitude to Reviewer **j89C** for providing valuable feedback. We are pleased to know that the reviewer acknowledges the intuitive and compelling motivation behind our work. We are encouraged by the reviewer's positive remarks, and we continue to address the reviewer’s concerns below:
>
> **Q30:** First of all, the main idea of ‘Type-to-Track’ has a similar idea to Referring Multi-Object Tracking [A] (released at Arxiv2023.03, accepted at CVPR2023), which both use language description to prompt multiple objects.
>
> **A30:** We appreciate Reviewer **j89C** for suggesting the Ref-KITTI dataset. Please refer to **KP2** and the attachment for the updated Table 1, and **A1** for our responsively class-agnostic tracking paradigm.
>
> **Q31:** I’m curious about the annotation details of the descriptions. Are they manually annotated? Or they are annotated by the current techniques of GPT?
>
> **A31:** All the descriptions are manually annotated by human annotators. The annotation process began before the release of the first GPT technique supporting visual input (GPT-4) on Mar 13th, 2023, and our first initiated annotation was already released on Feb 17th, 2023. Note that the current visual GPT techniques mainly focus on images or videos, while our task involves tracking objects, which demands **the temporal consistency of each instance** throughout the entire length of the videos. This distinction highlights the unique challenges of our task, which **can not be fully addressed by existing visual GPT techniques**.
>
> **Q32:** In Table 3, does the number of words represent the number of descriptions? Or the words are words of descriptions?
>
> **A32:** It is the number of words in descriptions. We counted other datasets in the same criteria.
>
> **Q33:** The evaluation of caption prompts confuses me. The crucial words of created captions have one-to-one object matching with the grounded objects. So each object is evaluated individually?
>
> **A33:** The term "one-to-one" in the evaluation of caption prompts refers to the fact that **each caption is matched to one specific object**. However, it **does not mean that objects are evaluated one-by-one**. In the evaluation process, all text prompts are **passed into a single network pass**, and the performance is measured by the **multiple object tracking metrics**. These MOT metrics **heavily penalize the inaccuracies in identity assignment**, especially as the number of objects increases, and **can not be evaluated individually**.
>
> **Q34:** This paper is not easy to follow and sometimes overclaimed on some details, e.g., third-order tensor modeling (using three inputs), and correlation simplification (using separate attention). Besides, the authors define many new concepts but lack details and explanations, e.g., the single-stage end-to-end framework. How to deal with new entrance or exit objects?
>
> **A34:** We appreciate your concerns and have taken them into account. Allow us to clarify the points you mentioned:
>
> - Third-order tensor modeling: The use of a third-order tensor refers to our specific approach to handle three types of input components (video frames, textual prompts, and object features) in a single tracker network. **A third-order tensor [E]** refers to **a three-dimensional tensor** that allows us to model the interactions and dependencies between these components effectively.
>
> - Correlation simplification: The correlation simplification process is detailed in the derivation from Eqn. 3 to Eqn. 6, where we show **the step-by-step development of the simplified correlation approach**. It involves streamlining the correlation computation to enhance efficiency and performance.
>
> - Single-stage end-to-end framework: The term "single-stage end-to-end framework" in our paper refers to **the joint detection-and-tracking approach that is fully trainable**. We compared this framework with **another two-stage framework, where each stage needs to be trained separately**.
>
> - New entrance or exit objects: In the Supplementary material, we provided a detailed description of how we handle such cases. For new entrance objects, the initialization process occurs from **line 13 to line 14** of our algorithm. When a new object appears in the video, we create a new tracklet for it and initialize its features based on the first appearance. On the other hand, for exit objects, we handle deprecations by removing inactive tracklets from further consideration which exceed a tolerant window time, as described in **line 24** of the algorithm.
>
> **Q35:** How about the visualization performance of the proposed method?
>
> **A35:** The visualization performance of our proposed method is presented in the supplementary material, where we provide a video demonstration and visualizations to showcase the accuracy, effectiveness, and failed cases of our approach. Some more cases are in the attachment.
>
> [E] Kolda, Tamara G., and Brett W. Bader. "Tensor decompositions and applications." SIAM review 51.3 (2009): 455-500.

---

> > ### Comment · Reviewer_j89C · 2023-08-20
> >
> > Overall, the rebuttal from the author side solves most of my concerns. So I decide to keep my original rating for it.
> >
> > I would encourage the authors to conduct a comparison&discussion with RMOT in the revised version, since RMOT (developed in the last year) should be much earlier than this work.
> >
> > Moreover, I notice that other reviewers mentioned about the quality of annotation. I would also suggest the authors to address since it is really a big bet for this work.

---

> > > ### Author Response · Authors · 2023-08-20
> > >
> > > Dear Reviewer **j89C**,
> > >
> > > We deeply appreciate your constructive comments.
> > >
> > > We already had the RMOT's dataset (Ref-KITTI) in comparison in Table 1 in the rebuttal PDF file. We also added the discussion in the main text accordingly. We will try to add the performance comparison.
> > >
> > > We have released ALL training annotations of version v1.0, including the MOT17 and TAO annotations.
> > >
> > > We welcome any further suggestions you may have, as we always strive to enhance our work's quality and impact.
> > >
> > > Best regards,
> > >
> > > Authors

---

### Official Review · Reviewer_RiBQ · 2023-07-04

**Soundness:** 3 good
**Presentation:** 4 excellent
**Contribution:** 3 good
**Rating:** 6
**Confidence:** 4

**Summary:**

This paper systematically tackles the prompt-based multiple object tracking problem by (1) introducing a dataset for the Grounding Multiple Object Tracking Task (GroOT), (2) introducing a new tacking paradigm (type to track), and (3) developing a new language-based multiple object tracking model (MENDER).

**Strengths:**

This paper systematically tackles the prompt-based multiple object tracking problem with a variety of relevant contributions. We list the strengths of the paper below, expanding each paragraph with comments on the relevance of each contribution.

## A new tracking paradigm (type to track)
This paper introduces the type to track paradigm, which consists in prompting models with text descriptions and obtaining all the tracklets corresponding to it. This paradigm is of great practical importance, as it allows users to query videos with text prompts to retrieve the desired objects.

## A dataset for the Grounding Multiple Object Tracking Task (GroOT)
This paper introduces GroOT, a dataset which extends MOT17 and TAO with a variety of text captions to learn models suitable for the type to track task.

GroOT supports 5 settings, including tracking by :
- category name
- category synonyms
- category definition
- retrieval prompts: 3 retrieval prompts per video are provided. the prompts are sequential and each prompt could refer to multiple tracklets at the same time (one-to-many).
- tracklet captions: one caption here corresponds to one tracklet, where the caption consistently describes the object throughout the tracklet. In the TAO subset, the caption describes the object appearance. In the MOT17 subset, 2 types of captions are provided for each object, one describing the action and the other the appearance.

## Class-agnostic evaluation metrics
To evaluate the type to track performance on GroOT, this paper introduces CA-MOTA and CA-IDF1, two class-agnostic tracking metrics.

## A new language-based multiple object tracking model (MENDER)
The authors also introduce MENDER, a proof-of-concept tracker capable of retrieving tracklets based on text prompts.

MENDER is composed of an image encoder, a text encoder, and a transformer decoder.
The **image encoder** encodes in parallel the image at time $t$ and the cropped bounding boxes for tracked instances at $t-1$.
The **text encoder** is a RoBERTa encoder that encodes the text prompts.
Cross-attention between image features, track embeddings and text embeddings produces the inputs to the transformer decoder, which is trained with a GIoU loss to detect the objects based on the track and text queries.
The proposed model is simple enough and addresses the type to track task effectively.

**Weaknesses:**

## Missing baselines
The `category_name` setting could be tackled by any multiple object tracking algorithm. The current version of the paper lacks an absolute estimate of the tracking performance of MENDER in comparison to state-of-the-art tracking algorithms. It is possible that the current design produces suboptimal tracking performance, and only empirical results can show whether this is true or not. The paper would thus benefit from a comparison of MENDER to SOTA tracking-by-detection methods on the `category_name` setting.

## Class-agnostic evaluation metrics
**Evaluation of retrieval performance.** The proposed metrics focus more on evaluating the tracking performance of retrieved tracklets rather than the retrieval performance itself. Although the limitations section already indicates this as a limitation, I find this to be a weakness since a benchmark for tracking retrieval should specifically provide suitable metrics for the evaluation of such task.

**CA-HOTA.** Although MOTA and IDF1 are established tracking metrics, the community has reached an agreement that they cannot solely represent the performance of a tracker. The HOTA metrics (DetA, AssA, and HOTA) disentangle different components of the tracking pipeline and it would be beneficial for the proposed benchmark to also introduce a class-agnostic version of such metrics.


## Missing ablation on the tokenizer and the text encoder
The authors adopt the RoBERTa tokenizer and text encoder. However, they do not provide an explanation for choosing it over other popular architectures, e.g. CLIP. The paper would benefit from a comparison to other text encoder architectures and initializations.


**Questions:**

I find the paper well-written and relevant to the community. However, the weaknesses listed above somewhat hinder the contributions of this paper.

In my opinion:
1. a comparison on the `category_name` setting to SOTA tracking-by-detection methods on GroOT is essential for this paper
2. retrieval-specific metrics would complement the contribution of the dataset, which as of now does not allow directly assessing the retrieval task performance
3. the paper would benefit from ablations on the tokenizer choice.

I would consider increasing my rating and supporting this paper if my current concerns are addressed, in particular points 1 and 2.

**Limitations:**

The authors have adequately pointed out limitations of their work.

---

> ### Author Rebuttal · Authors · 2023-08-05
>
> We extend our sincere appreciation to Reviewer **RiBQ** for providing constructive feedback and recognition of our contributions. We are delighted to see that the reviewer has recognized the multiple relevant contributions of our paper. We will continue to refine and enhance our work per your suggestions.
>
> **Q26:** Missing baselines.
>
> **A26:** We appreciate Reviewer **RiBQ** for suggesting the performance comparison with SOTA approaches. The `category_name` is also the official benchmark, we published our result of the `category_name` setting on the official leaderboard of MOT17 (please find the method MENDER on the Private detection track, we are not allowed to attach any external links). It allows **convenient and direct comparison with other approaches**, including ByteTrack [B]. Note that the proposed MENDER is **one of the first attempts at the Grounded MOT task**, not to achieve the top rankings on the general MOT leaderboard. Other SOTA approaches benefit from the efficient single-category design in their separate object detectors, while our design is **agnostic to the category** and for **flexible textual input**. Below is the official comparison of some methods:
>
> |Method           |MOTA   |IDF1   |HOTA   |AssA   |DetA   |
> |-----------------|-------|-------|-------|-------|-------|
> |MENDER           |65.0   |67.1   |53.9   |54.4   |53.6   |
> |TrackFormer      |74.1   |68.0   |57.3   |54.1   |60.9   |
> |ByteTrack        |80.3   |77.3   |63.1   |62.0   |64.5   |
>
> **Q27:** Evaluation of retrieval performance.
>
> **A27:** As mentioned in the limitations section of our paper, this is indeed a recognized limitation of our work. We set a concrete goal in this work to lay the groundwork for future research in this emerging field and to inspire further investigations. As we continue to refine and expand the benchmark, we value the reviewer's feedback and we will explore additional metrics that can better capture the retrieval performance aspects.
>
> **Q28:** CA-HOTA
>
> **A28:** We appreciate Reviewer RiBQ for suggesting another class-agnostic metric for the comprehensive evaluation. As we stick with the idea of ignoring the negative effect of classification, we formulate the CA-HOTA as below:
>
>
> $\text{HOTA} = \frac{1}{|CLS|}\sum_{cls}^{CLS}\Big(\sqrt{DetA\cdot AssA}\Big)_{cls}$
>
> $\text{CA-HOTA} = \sqrt{(DetA_{CLS^1})\cdot(AssA_{CLS^1})}$, where the category set $CLS$ with size $n$ is reduced to $1$ by combining all elements: $CLS^n\rightarrow CLS^1$
>
> We also report the additional metrics namely CA-HOTA, AssA a DetA for the TAO dataset on the `tracklet_captions` setting as follows. We will organize and include them in other settings.
>
> |Method           |CA-HOTA   |AssA   |DetA   |
> |-----------------|----------|-------|-------|
> |MDETR + Tfm      |14.9      |19.3   |11.7   |
> |MENDER           |19.5      |26.0   |15.8   |
>
> **Q29:** Missing ablation on the tokenizer and the text encoder.
>
> **A29:** We appreciate Reviewer **RiBQ** for suggestions on the ablation of the text encoder. In our initial study, we opted for the RoBERTa tokenizer and text encoder based on its widespread usage and proven performance in various grounded baselines, including MDETR. We were initially constrained by the significant resources needed for further evaluations. As part of our commitment to continuously improve our work, we plan to conduct a comprehensive analysis of various text encoder architectures and initialization methods in our future iterations.

---

> > ### Comment · Reviewer_RiBQ · 2023-08-13
> >
> > I thank the authors for their reply and for having addressed some of my concerns. I recommend including the performance comparison of MENDER to other trackers in the main paper, and work on improving its absolute tracking performance.
> >
> > Additional comments on the dataset quality can be found in my reply to the general rebuttal. I will keep an open mind, but after careful inspection I have become very concerned by the quality of the annotations.

---

> > > ### Author Response · Authors · 2023-08-18
> > >
> > > Dear Reviewer **RiBQ**,
> > >
> > > We appreciate your invaluable efforts and insightful comments, which have helped improve our annotation quality. We have updated the MOT17 dataset to address your annotation concern. We would like to hear your feedback. Please feel free to suggest if you have other feedback.
> > >
> > > Best regards,
> > >
> > > Authors

---

> > > > ### Author Response · Authors · 2023-08-21
> > > >
> > > > Dear Reviewer **RiBQ**,
> > > >
> > > > We have addressed your concerns in our responses. We also completed and released version v1.0 training annotations of our GroOT dataset, including MOT and TAO subsets.
> > > >
> > > > Please feel free to raise questions if you have other concerns.
> > > >
> > > > Best regards,
> > > >
> > > > Authors

---

### Official Review · Reviewer_9rR3 · 2023-07-06

**Soundness:** 3 good
**Presentation:** 3 good
**Contribution:** 4 excellent
**Rating:** 7
**Confidence:** 4

**Summary:**

This paper delves into language-grounded multi-object tracking, a highly important task that has been underexplored due to the lack of suitable datasets.  Firstly, the authors propose a novel dataset that extends existing Multiple Object Tracking (MOT) datasets with additional captions describing appearance or action of each tracklet, as well as retrieval prompts describing main content of a scene. To evaluate language-grounded multi-object tracking, the authors consider five scenarios with different types of text input (prompts) and suggest class-agnostic MOTA and IDF1 that decouple the classification accuracy. Also, the authors introduce MENDER, an efficient single-stage model that autoregressively tracks objects relevant to the prompt. The effectiveness of this model is validated through a comparison with a two-stage baseline model consisting of MDETR and TrackFormer.

**Strengths:**

* This paper addresses a significant problem, aiming to retrieve multiple objects relevant to an input prompt from a video. This setting is novel compared to existing spatio-temporal video grounding methods (e.g., TubeDETR[1]), which are limited to retrieving only a single object. The proposed task can not only serve as a user-friendly video analysis application but also work as a video perception foundation model for fine-grained video understanding.
* The proposed dataset is significant to further research on video-language model.
* The proposed diverse scenarios for evaluating grounded MOT are interesting and valid.

[1] A. Yang, et al., TubeDETR: Spatio-Temporal Video Grounding with Transformers

**Weaknesses:**

## Inconsistent annotations
  * In the released dataset, some inconsistent annotations are observed. How quality assurance is achieved in the manual annotation process? Is there criteria to accept or reject the captions from the annotator?
  * In Sec 3.1, L116 states that the appearance-related caption and action-related caption are provided in order if the tracked object is person. However, in the annotations, the captions are randomly ordered. For example, in mot17_train file, object whose instance_id is 170 has captions of ["person walking on sidewalk", "person wearing black jacket"], where action-related caption is the first in the caption list.
  * Also, object whose instance_id is 9 has three captions: ["a person sitting on a bench", "child in stroller", "baby stroller"]. And, in TAO dataset, there are some instances with six captions. What is the reaons of such variance in caption annotations?
  * I would rather suggest to have separate field for appearance described caption and action described caption.

## Heavy dependence on TAO dataset
* MOT Challenge datasets include more challenging tracking cases caused by complex motion and occlusion, compared to TAO dataset. However, its scale is very small (e.g., only 7 videos for training).
* I think captioning other human-centric MOT datasets with complex motion would improve the importance of the dataset. For example, human-centric datasets with dense FPS video (e.g., DanceTrack[2] and HiEve[3]) are the possible candidates.

## Unclear evaluation metrics
* In the paper, I cannot find the details of matching GT-prediction pairs which are necessary to compute FN and FP.
* What is the reason of cls exists in Eq. 1 even though we are not considering class of tracklets? Isn't that the class of tracklet do not exist in the case of grounded MOT?

## Lack of experimental details
* Unclear details about implementing the two-stage baseline model which is based on the MDETR and TrackFormer. TrackFormer jointly performs object detection and tracking, but how MDETR is integrated into TrackFormer to achieve grounded object detection?

[2] P. Sun, et al., DanceTrack: Multi-Object Tracking in Uniform Appearance and Diverse Motion, CVPR 2022
[3] W. Lin, et al., Human in events: A large-scale benchmark for humancentric video analysis in complex events, arxiv 2020.05

**Questions:**

Major concerns are raised in the weakness section.

Here are some minor questions:
* Why only text prompt for every frame exists, but there is no prompt-tracklet paired annotations in the MOT files?
* In Table 3, dataset statistics for MOT17 videos are not correct. As far as I know, there are 7 unique videos for the training and test subsets, respectively. Also, your released dataset seems using the GT tracklets from the unique videos.
* How do you have the mot17_test annotation file? Is the tracklet obtained from GT or just prediction value?
* TAO dataset is the most important one as the proposed dataset is built on top of this one, but why it is not included in Table 1?
* Why do you design the MENDER as online tracking model that runs autoregressively, instead of predicting multiple spatio-temporal tubes at once?
* In Table 4 and 5, is mAP the absolute value or percentage value?
* In Table 4, some bolded values are not the best in the metric.

**Limitations:**

The limitations and future works are well addressed.

---

> ### Author Rebuttal · Authors · 2023-08-05
>
> We extend our gratitude to Reviewer **9rR3** for your thoughtful and valuable feedback. We are pleased to see that the significance of our work has been acknowledged.
>
> **Q12:** How quality assurance is achieved in the manual annotation process? Is there criteria to accept or reject the captions from the annotator?
>
> **A12:** Similar to [D] presented in NeurIPS 2022 Datasets and Benchmarks, during the annotation process, we conduct regular quality reviews by batch to identify any errors. Regarding acceptance or rejection, captions are evaluated based on the cross-checked voting by all other annotators. We provide clear guidelines and instructions to the annotators to choose **the most distinct feature of the targets in the videos**.
>
> **Q13:** What is the reaons of such variance in caption annotations?
>
> **A13:** Note that the version v0.1 we released on Feb 17th, 2023 **long ago before the submission deadline**. Therefore it is just an initial version for testing the annotation procedure and before the cross-checked step. We published a newer version v0.2 on Jul 8th, 2023 (before this rebuttal) which has solved some inconsistent instances. The data demonstration and results are shown at the version v0.6. The first official ready version, v1.0, is currently under assessment to ensure the quality and consistency of the annotations. We ensure the maintenance of the annotations in the long term, and we will cooperate with the MOTChallenge organizers upon this paper’s acceptance and their approval.
>
> **Q14:** I would rather suggest to have separate field for appearance described caption and action described caption.
>
> **A14:** We appreciate your suggestion. We will consider organizing the data format in this way.
>
> **Q15:** Heavy dependence on TAO dataset
>
> **A15:** We acknowledge **the unique characteristics of each dataset**, including the wide range of categories in TAO, the distinct human appearance in MOT17, and the dense population in MOT20. Although the MOT17 and MOT20 have a limited scale, we decided to combine all datasets to **diversify our scenarios**. We appreciate the reviewer’s suggestion on captioning DanceTrack and HiEve datasets. We will consider contributing to these aspects in future work.
>
> **Q16:** In the paper, I cannot find the details of matching GT-prediction pairs which are necessary to compute FN and FP.
>
> **A16:** We followed the original formulas of MOTA and IDF1 for all the components FN, FP, IDS, IDTP, IDFP, and IDFN of the metrics. The score threshold required for a TP match is 0.5, following the TrackEval implementation.
>
> **Q17:** What is the reason of cls exists in Eq. 1 even though we are not considering class of tracklets?
>
> **A17:** Originally, we derived the per-category form of MOTA and IDF1 and their class-agnostic form (i.e. CA-MOTA and CA-IDF1), and included the $cls$ notation to show the difference in using category. However, we had to remove the original forms due to the page limit and added a description to explain the $cls$ notation. In the revision upon acceptance, we will show the derivation of both forms as below:
>
> $\text{MOTA} = \frac{1}{|CLS|}\sum_{cls}^{CLS}\Big(1 - \frac{\sum_{t}{(FN_{t} + FP_{t} + IDS_{t})}}{\sum_{t}{GT_{t}}}\Big)_{cls}$
>
> $\text{CA-MOTA} = 1 - \frac{\sum_t\sum_{cls}{(FN_{t} + FP_{t} + IDS_{t})}}{\sum_t(\sum_{cls}{GT_{cls}})_{t}}$
>
> $\text{IDF1} = \frac{1}{|CLS|}\sum_{cls}^{CLS}\Big(\frac{2\times IDTP}{2\times IDTP + IDFP + IDFN}\Big)_{cls}$
>
> $\text{CA-IDF1} = \frac{\sum_{cls}{(2\times IDTP)}}{\sum_{cls}{(2\times IDTP + IDFP + IDFN)}}$
>
> **Q18:** Unclear details about implementing the two-stage baseline model which is based on the MDETR and TrackFormer.
>
> **A18:** The TrackFormer’s source code has a mode to receive public detections as input. We generated grounded detections from MDETR and saved them to a compatible format, then imported them to the public detection mode of TrackFormer.
>
> **Q19:** Why only text prompt for every frame exists, but there is no prompt-tracklet paired annotations in the MOT files?
>
> **A19:** We follow the original design of the TAO dataset's annotations for the consistency of our annotation system working on two datasets. That decision was made to maintain compatibility and consistency and ensure simplicity and ease of use, enabling users to leverage a wide range of existing COCO API readers for data analysis and processing.
>
> **Q20:** In Table 3, dataset statistics for MOT17 videos are not correct.
>
> **A20:** Thank you for pointing this out. We corrected Table 3 and also the statistics in Table 1 in the attachment.
>
> **Q21:** How do you have the mot17_test annotation file? Is the tracklet obtained from GT or just prediction value?
>
> **A21:** The tracklet identities are the prediction value. Please see **KP3**.
>
> **Q22:** TAO dataset is the most important one as the proposed dataset is built on top of this one, but why it is not included in Table 1?
>
> **A22:** We did have the TAO dataset’s statistics in Table 1 and Table 3 in the first version, but we removed it in Table 1 due to the page limit. Please refer to the updated Table 1 in the attachment. We will re-add the TAO dataset in the revision upon acceptance.
>
> **Q23:** Why do you design the MENDER as online tracking model that runs autoregressively, instead of predicting multiple spatio-temporal tubes at once?
>
> **A23:** Online is a preferred practice as widely accepted by the community specifically for the tracking problem. We will add this discussion of the online tracking practice compared with the spatiotemporal tube practice defined by TubeDETR.
>
> **Q24:** In Table 4 and 5, is mAP the absolute value or percentage value?
>
> **A24:** It is the absolute value.
>
> **Q25:** In Table 4, some bolded values are not the best in the metric.
>
> **A25:** Thank you for pointing out the typos. We corrected them in Table 4.
>
> [D] Doersch, Carl, et al. "Tap-vid: A benchmark for tracking any point in a video." Advances in Neural Information Processing Systems 35 (2022): 13610-13626.

---

> > ### Comment · Reviewer_9rR3 · 2023-08-14
> > **Regarding the Response**
> >
> > ## Overall response
> > Thank you for clarifying my concerns. I agree that the proposed dataset is significant at this stage for researching multi-modal MOT and the proposed task is also interesting. However, the current version of the dataset (only v0.2 is available to the public on 2023-08-14) is not yet satisfying and I would like to keep my score.
> >
> > ## Remaining concern about annotations
> > Please note that as the v0.2 dataset is only available at this moment, my review is based on that.
> >
> > ### About captions describing the human action
> > The captions are restricted to describing the appearance of an object or person, instead of action. In the MOT17 train set, there are only 14 words related to verb - filtered by NLTK - ['carrying', 'crossing', 'eating', 'holding', 'is', 'parked', 'pushing', 'riding', 'sitting', 'standing', 'striped', 'talking', 'walking', 'wearing']. Among these words, only some words are related to describing the actions of a person. Based on this observation, I cannot agree with your contribution point (L8) "textual captions describing their **appearance and action** in detail" which was appealing to me.
> >
> > ### Limited diversity of captions
> > In MOT17 train set, among 175540 captions, there are 294 unique captions and its distribution is severely long-tailed, where top-5 frequent captions occupy 22.6%. Even such top-5 frequent captions contain the same meaning (1, 2, 4) and their description is not informative as well. I guess this is caused by the characteristics of the MOT17 dataset which focuses on the scene including pedestrians. However, I believe this would be handled effectively by carefully annotating the captions (e.g., describing the fine-grained appearance) or separately annotating the captions about appearance and action (e.g., walking or standing).
> > 1. 'a person walking on the sidewalk'
> > 2. 'person walking on the sidewalk'
> > 3. 'person standing on sidewalk'
> > 4. 'person walking on sidewalk'
> > 5. 'person wearing black shirt'

---

> > > ### Author Response · Authors · 2023-08-18
> > >
> > > Dear Reviewer **9rR3**,
> > >
> > > We appreciate your invaluable efforts and insightful comments, which have helped improve our annotation quality. We have updated the MOT17 dataset to address your annotation concern. We would like to hear your feedback. Please feel free to suggest if you have other feedback.
> > >
> > > Best regards,
> > >
> > > Authors

---

> > > > ### Comment · Reviewer_9rR3 · 2023-08-18
> > > >
> > > > My concern is well addressed in the v1.0 MOT17 dataset and I am willing to raise my score. It would be great to include the analysis of captions that are mentioned before, such as frequency/distribution of words - verb (action) / noun (type of object) / adjective (attribute of object) - used in the captions. I hope to see the updated version of the TAO dataset soon.

---

> > > > > ### Author Response · Authors · 2023-08-22
> > > > >
> > > > > Dear Reviewer **9rR3**,
> > > > >
> > > > > We deeply appreciate your constructive comments, your engagement during the discussion, and your support.
> > > > >
> > > > > We have released ALL training annotations of version v1.0, including the MOT17 and TAO annotations.
> > > > >
> > > > > Best regards,
> > > > >
> > > > > Authors

---

### Official Review · Reviewer_h9ub · 2023-07-06

**Soundness:** 3 good
**Presentation:** 3 good
**Contribution:** 3 good
**Rating:** 7
**Confidence:** 4

**Summary:**

The paper introduces a novel task and dataset that involves tracking multiple objects in videos using language prompts. It proposes two evaluation protocols: retrieval prompts, where the prompts can change during the tracking process, and caption prompts, which remain consistent. The paper also presents class-agnostic evaluation metrics by modifying MOTA and IDF1. Additionally, a novel framework called MENDER is introduced to address the grounded MOT task.

**Strengths:**

1. The significance of the proposed task and dataset is well-defined, as existing tracking datasets using language prompts have been focused on single-object tracking (SOT) tasks.
2. The evaluation protocol for the proposed dataset consists of five distinct protocols, each with clear explanations of their purpose. This dataset stands out from previous tracking datasets due to its diverse range of prompts.
3. The proposed MENDER framework demonstrates efficient and effective performance in addressing the grounded MOT task.
4. The paper is overall well-written and easy to follow.

**Weaknesses:**

1. Referring video segmentation datasets like Ref-YTVOS and Ref-DAVIS are also related to this grounded MOT task since they also aim at tracking a referred target in a video. I think these datasets should also be mentioned in the paper as a comparison benchmark.
2. Moreover, I think Refer-KITTI [1] could also be discussed in the paper as it also utilizes referring expressions for multi-object tracking.
3. The experimental comparisons in the newly proposed dataset are limited. It would be beneficial to include additional methods such as TransRMOT [1] and popular MOT baselines like ByteTrack [2] for a more comprehensive evaluation of the dataset's performance. Adding these comparisons would provide stronger validation for the dataset.

[1] Referring Multi-Object Tracking, https://arxiv.org/pdf/2303.03366.pdf

[2] ByteTrack: Multi-object tracking by associating every detection box, ECCV 2022

**Questions:**

As discussed in the weaknesses, I hope the authors would supplement some dataset comparisons and experimental validations.

**Limitations:**

Yes, the authors have adequately addressed the limitations of their work.

---

> ### Author Rebuttal · Authors · 2023-08-05
>
> We sincerely thank Reviewer **h9ub** for your thoughtful and constructive comments. We are pleased to see that the significance of our proposed task and dataset has been acknowledged. Furthermore, we are glad to receive positive feedback on the efficiency and effectiveness of our proposed MENDER framework. We are grateful for the reviewer's valuable input, and we will carefully consider all suggestions to enhance the quality in the upcoming revision.
>
> **Q9:** Referring video segmentation datasets like Ref-YTVOS and Ref-DAVIS are also related to this grounded MOT task since they also aim at tracking a referred target in a video.
>
> **A9:** Please refer to **KP1** and **A2**. We added them to the comparison in Table 1 in the attachment.
>
> **Q10:** Moreover, I think Refer-KITTI [A] could also be discussed in the paper as it also utilizes referring expressions for multi-object tracking.
>
> **A10:** Please refer to **KP2**. We added them to the comparison in Table 1 in the attachment.
>
> **Q11:** The experimental comparisons in the newly proposed dataset are limited.
>
> **A11:** We appreciate Reviewer **h9ub** for suggesting another comprehensive comparison. We submitted our prediction to the official leaderboard for a convenient and direct comparison with other SOTA approaches on the `category_name` setting. We will try to test SOTA’s performance on other settings. Please refer to **A26**.

---

> > ### Comment · Reviewer_h9ub · 2023-08-14
> > **About the comparison with other approaches in catogory_name setting**
> >
> > I would like to express my appreciation to the authors for providing a rebuttal and for addressing some of my concerns. However, my concern about the comparison with other baselines is not fully resolved.
> >
> > As suggested in A26, the performance of MENDER on `category_name` setting is far behind the SOTA tracker TrackFormer and ByteTrack. It is understandable that ByteTrack's superior performance could be achieved from their task-specific design for associations. However, it is hard to find reasons why MENDER's performance falls short compared to TrackFormer, which is essentially a simple extension of Deformable DETR.
> >
> > Considering that MENDER benefits from additional training with more language descriptions, one would naturally expect it to gain advantages from the increased annotations. Rather, the result in A26 rather gives an impression that the proposed method suffers from generalizing to a more standard `category_name` setting.

---

> > > ### Author Response · Authors · 2023-08-14
> > > **A reminder on the fair comparison**
> > >
> > > We would like to remind Reviewer **h9ub** that compared to Trackformer, MENDER only demonstrates a **marginal decrease in identity assignment** (67.1 vs 68.0 IDF1 and 53.9 vs 57.3 HOTA). The distinction in the MOTA detection metric stems from our detector's design, which is a Grounded detector integrating prompts as an additional input. Fair comparisons with the two-stage Trackformer were already provided in the main paper on four other settings. Below is an additional comparison of the `category_name` setting:
> > >
> > > |Method           |MOTA    |IDF1   |HOTA   |mAP     |
> > > |-----------------|--------|-------|-------|--------|
> > > |TrackFormer      |64.7    |65.8   |56.9   |0.793   |
> > > |MENDER           |67.0    |71.2   |61.4   |0.876   |

---

> > > > ### Comment · Reviewer_h9ub · 2023-08-21
> > > >
> > > > Thanks for your response. Although the absolute performance of MENDER falls short of other SOTA MOT models, its role as an example baseline for grounded MOT is somewhat sufficient. Furthermore, I've carefully gone through other reviewers' opinions about the dataset quality and think the problem has been sufficiently addressed in the final version of the dataset. Overall, I think this dataset could positively impact the MOT community for grounded object reasoning. Thus I'd like to keep my original rating and vote for accepting the paper. But I recommend the authors to keep working on improving the absolute performance of MENDER.

---

### Official Review · Reviewer_HDeT · 2023-07-06

**Soundness:** 3 good
**Presentation:** 2 fair
**Contribution:** 3 good
**Rating:** 5
**Confidence:** 4

**Summary:**


The authors have created a new dataset by combining existing Tracking datasets and adding language prompt annotations. They also proposed a method by extending and modifying MDETR and Trackformer.


**Strengths:**

1. The dataset quality looks sound, diverse, and sufficiently large enough to train Deep networks.
2. The dataset is well documented and the task is clearly defined.
3. The proposed solution is simple and should be easy to extend by new researchers.


**Weaknesses:**

1. The authors claim that Type-to-Track is a novel paradigm. This is incorrect as there exists Datasets, e.g., Refer-Youtube-VOS, Ref-DAVIS, Ref-KITTI, having the same challenge.
2. The related work in incomplete. The authors fail to compare their proposed dataset with existing datasets from Video Instance Segmentation(VIS) literature. Note that, MOTS and VIS are the same task with different nomenclature.
3. The modeling part is unnecessarily written in a complicated way. It is a simple extension of DETR architecture to model the specific solution. Instead of complicating the method with notation, The authors should try to simplify the method section.
4. There are too many evaluation metrics. There are 5 different sub-categories of the same task with separate evaluations on the 3 base datasets. So there are in total, around 12*6=72 different evaluation metrics.

**Questions:**

1. How is this dataset different from Refer-Youtube-VOS, Ref-DAVIS, and Ref-KITTI?
2. How did you evaluate MOT17 and MOT20 datasets? The test ground truth is not public for these datasets. What is MOT17 subset in Tab. 4 and 5?
3. Why is there  no unifying evaluation metric. The authors report evaluation separately on all the different datasets they worked on. A new dataset needs to be either compatible with previous ones, or propose a new metric. Without a single metric, new line of researchers would find it difficult to adapt the dataset.
4. How many human hours was needed to annotate the dataset?








**Limitations:**

yes

---

> ### Author Rebuttal · Authors · 2023-08-05
>
> We sincerely thank Reviewer **HDeT** for your valuable and thoughtful comments. We are glad that the Reviewer agrees with the positive aspects of our work, including the acknowledgment of the dataset's quality through documentation and the clear definition of the task. Moreover, we are pleased to know that the proposed solution is perceived as simple and easily extendable. We will take all suggestions into careful consideration in our revision.
>
> **Q1:** The authors claim that Type-to-Track is a novel paradigm. This is incorrect as there exists Datasets, e.g., Refer-Youtube-VOS, Ref-DAVIS, Ref-KITTI, having the same challenge.
>
> **A1:** Our proposed Type-to-Track paradigm is distinct in its focus on **responsive and conversational** typing to track **any objects** in videos, requiring **maintaining the temporal motions of multiple objects of interest**. In contrast, the Video Object Segmentation (VOS) challenge **primarily measures the overlapping area between the ground truth and prediction** for **a single foreground object in each caption**, with less emphasis on densely tracking multiple objects over time. And the Ref-KITTI [A] challenge is **not agnostic to the category** and **not able to change the prompt responsively**. Therefore, our statement on the novel paradigm is still valid, as stated in lines 35-36 in the main paper.
>
> **Q2**: The authors fail to compare their proposed dataset with existing datasets from Video Instance Segmentation(VIS) literature.
>
> **A2**: We acknowledge the importance of comparing our proposed dataset with existing datasets from the Referring Video Object Segmentation literature (Refer-Youtube-VOS, Ref-DAVIS), which involves **a single object along with a single referring expression**, in the same category as Single Object Tracking (SOT). The mentioned VIS task from the Reviewer may be a typo that typically deals with multiple objects without considering Referring Expressions. We appreciate your feedback and we added these two datasets to the comparison in Table 1.
>
> **Q3:** The modeling part is unnecessarily written in a complicated way. It is a simple extension of DETR architecture to model the specific solution.
>
> **A3:** We want to emphasize that our method **goes beyond a simple extension and presents an efficient solution to the tracking problem**. Our approach is specifically designed to address **the complex problem of multi-modality**, where we handle three types of input components in a single tracker network. It involves modeling the interactions between video frames, textual prompts, and object features, which is **a unique challenge not addressed by standard DETR**. In our paper, we take great care to present **a clear and detailed formulation of our approach**. We provide **step-by-step explanations** in both mathematical formulas (from Eqn. 2 to Eqn. 3) and visual representations to help readers **understand the motivation and the key components of our method** (Eqn. 5 and Fig. 4a), **enabling the audience can reproduce our algorithm and conduct further research** in this area.
>
> **Q4:** There are too many evaluation metrics.
>
> **A4:** The reason for employing different evaluation metrics is to **ensure a comprehensive assessment** of the proposed method. Our decision to employ a range of evaluation metrics is also driven by **the nature of the problem and the standard protocol for tracking tasks**. As pointed out by Reviewer **RiBQ**, relying solely on metrics like MOTA and IDF1 can not fully capture the performance of a tracker. Similarly, the 5 different sub-categories explore various settings of the prompt, as detailed in our ablation study.
>
> **Q5:** How is this dataset different from Refer-Youtube-VOS, Ref-DAVIS, and Ref-KITTI?
>
> **A5:** Please refer to **KP1**, **KP2** and **A1**. As also indicated by Reviewer **h9ub**, our dataset stands out from previous tracking datasets due to **its diverse range of prompts**.
>
> **Q6:** How did you evaluate MOT17 and MOT20 datasets? What is MOT17 subset in Tab. 4 and 5?
>
> **A6:** The MOT17 subset in Table 4 and Table 5 is evaluated on the sub-optimal test set ground truth. Please refer to **KP3**.
>
> **Q7:** Why is there no unifying evaluation metric.
>
> **A7:** Please refer to **A4**.
>
> **Q8:** How many human hours was needed to annotate the dataset?
>
> **A8:** To ensure high-quality annotations, we engaged a team of ten professional annotators, each dedicating 15 hours per week to the task. The annotation work commenced in January and continued until the end of July. Haft of the time is spent to be cross-checked manually to maintain accuracy and consistency. As of now, the final version of the dataset is under assessment to ensure that it meets all the necessary criteria and adheres to the established quality standards.

---

> > ### Author Response · Authors · 2023-08-21
> >
> > Dear Reviewer **HDeT**,
> >
> > We have addressed your concerns in our responses. Please feel free to raise questions if you have other concerns.
> >
> > Best regards,
> >
> > Authors

---

> > > ### Comment · Reviewer_HDeT · 2023-08-22
> > >
> > > Thank you for your response. As many reviewers raised concerns about the dataset quality, I took a closer look. It has inconsistent and other issues as raised by other reviewers. Also, I feel that the overall complex presentation of the method and various matrices are somewhat confusing. Thus I will keep my rating.

---

### Author Rebuttal · Authors · 2023-08-05

We thank all reviewers for their valuable comments and suggestions. The reviewers highlight the strong aspects of the paper, including the high-quality and diverse dataset, the well-defined task, and the paper's clarity and organization. Reviewers **h9ub** and **RiBQ** are inclining towards acceptance (**Accept** and **Weak Accept**) due to the task's significance and the paper's well-structured presentation. Similarly, Reviewers **HDeT** and **j89C** lean towards acceptance with **two Borderline Accept** ratings. In contrast, Reviewer **9rR3** has assigned a **Borderline Reject** rating, mainly stressing the need for some additional details. We will address the common key points (**KP**s) first. Then the individual answers will be given in the following comments.

**KP1: Dataset comparison with Video Object Segmentation (VOS):** We thank reviewers **HDeT** and **h9ub** for suggesting Refer-Youtube-VOS, and Ref-DAVIS datasets. We added these two datasets to the comparison Table 1 with the specification of the VOS task and handling **a single object for each expression**. However, we want to clarify the importance and difference that makes our dataset stand out from these two datasets is the ability to construct the Grounded MOT task in **five different types of setting**, including **a setting that covers multiple objects in a single expression**. These five settings give us the full capability of evaluating different input types, which is also agreed upon by Reviewer **h9ub**.

**KP2: Dataset comparison with Ref-KITTI:** We thank reviewers **HDeT**, **h9ub**, and **j89C** for suggesting the Ref-KITTI [A] dataset. We included that dataset in Table 1. Note that our dataset focuses on **responsively tracking**, **agnostic to category**, and currently **outnumbers the frames and settings**, while the Ref-KITTI contains only two categories which are cars and pedestrians. At the time we submitted this paper on May 11th, 2023, we were unaware of the availability of the Ref-KITTI dataset. According to the comparison policy of NeurIPS 2023, we consider the Ref-KITTI [A] dataset as a concurrent work to our submission because the first time it appeared is Mar 11th, 2023 on arXiv and officially on proceedings of CVPR 2023 is June 2023.

Please find the updated Table 1 in the attachment. We will discuss the VOS task (Refer-Youtube-VOS, Ref-DAVIS) and the Ref-KITTI in the main text of the revision accordingly.

**KP3: Ours MOT17 test annotation is a sub-optimal ground truth:** We appreciate reviewers **HDeT** and **9rR3**’s inquiry, as it allows us to provide additional details to our experiments. We annotated the raw tracking data of the best-performant tracker at the time we constructed experiments as the sub-optimal ground truth (i.e., BoT-SORT [C] at 80.5% MOTA and 80.2% IDF1). That is also the raw data we used to evaluate all our ablation studies. The MOT17 subset in Table 4 on the `category_name` setting has **a similar performance** to our official result (please find the method MENDER on the Private detection track of the MOT17 challenge, we are not allowed to attach any external links): 67.00% MOTA and 71.20% IDF1 vs. 65.0% MOTA and 67.1% IDF1. We would cooperate with the MOTChallenge organizers for the official test prompts upon their approval. We had this detail in the main paper in the first version but removed it due to the page limit. We will re-add this detail to the revision upon acceptance.

[A] Wu, Dongming, et al. "Referring Multi-Object Tracking." CVPR 2023.

[B] Zhang, Yifu, et al. "Bytetrack: Multi-object tracking by associating every detection box." ECCV 2022.

[C] Aharon, Nir, Roy Orfaig, and Ben-Zion Bobrovsky. "BoT-SORT: Robust associations multi-pedestrian tracking." arXiv preprint arXiv:2206.14651 (2022).

---

> ### Comment · Reviewer_RiBQ · 2023-08-13
> **Poor dataset quality**
>
> I thank the authors for the rebuttal, and for addressing some of my concerns. I was originally positive towards this submission, I think that the community will greatly benefit from an MOT dataset with carefully-curated language captions and from a method to solve the type-to-track problem.
>
> After a more careful inspection of the dataset, I have however changed my mind. My main concerns are that: (i) the quality of the annotations is extremely poor; (ii) the annotation policy is inconsistent; (iii) the authors claim to have hired a team of ten professional annotators, but the quality of the captions seem to point out that a visual-language model was used for captioning; (iv) the proposed tracking algorithm trained on GroOT performs significantly worse than industry standards such as ByteTrack.
>
> - **Underwhelming performance of MENDER.** Regarding (iv), refer to the authors reply A26. While I appreciate that they disclosed the performance of MENDER, it also shows that the proposed algorithm is not particularly effective.
>
> - **Lack of retrieval metrics.** I believe that a paper introducing a new task (type-to-track) and a new dataset (GroOT) should provide metrics to evaluate the effectiveness of a given method at solving the type-to-track task. Since type-to-track is a retrieval task, I find it hard to accept a paper that does not introduce metrics to evaluate the retrieval performance.
>
> - **Poor quality of annotations.** I will here report some examples based on the supplementary video shared by the authors themselves.
>
> 1. _Timestamp 1:50._ inconsistent annotations in the same frame.
>
> 1.a) Three very similar bikes are described as "bicycle", "a black bike", "two black wheels on a bike". First of all, all bikes are black. I wonder how one could evaluate type-to-track performance (i.e. retrieval) if descriptions do not follow a common pattern. Second, "two black wheels on a bike" is an odd description and is the first example that reminds of captions from visual-language models.
>
> 1.b) "A man riding a bicycle", "a man wearing a helmet", "a person in a pink turban". They are all riding a bicycle, and all should be retrieved by the prompt a man riding a bicycle. Yet, proper retrieval metrics would penalize retrieval of "a man wearing a helmet" and "a person in a pink turban" when using such a prompt. Using well-defined patterns such as "a man riding a bicycle and wearing a helmet" and "a person riding a bicycle in a pink turban" would assist the evaluation of type-to-track methods.
>
> 2. _Timestamp 1:58._  Object ID 5274 was annotated as "the headlight of a car". I wonder how a human annotator could come up with this kind of description for what is seemingly a black car.
>
> 3. _Timestamp 2:13._ The authors say that in MOT they provide two distinct set of captions, i.e. appearance captions and action captions. However, the annotations shown here are really confusing and inconsistent. In the same frame, appearance captions sometime represent actions, e.g. some objects are labeled as "a man walking in a store", "woman walking on a sidewalk", "woman walking down the sidewalk" and others like "a man wearing a black shirt", "a little girl in a green shirt and white pants", "a man wearing a white hat".
>
>
> 4. _Timestamp 2:25._  The "man talking on a cell phone" is actually just touching his chin.
>
>
> Overall, my greatest concern is the **underwhelming quality of the annotations**. A by-product of the poor quality of the annotations is that it is really hard to evaluate the performance of the type-to-track task. In my opinion, a dataset designed for a task should provide all the tools to evaluate such task, while these annotations make it impossible.
>
> While there is no right way to annotate visual-language datasets, this dataset does not seem to be going in the right direction. In my opinion, the dataset would need significant further refinement before getting published.  However, I am waiting to hear the opinion of other reviewers and keep my mind open to different opinions.

---

> > ### Author Response · Authors · 2023-08-14
> > **Response to Reviewer RiBQ's comments**
> >
> > We thank Reviewer **RiBQ** for the constructive comments. Allow us to clarify the points you mentioned:
> >
> > - Quality of annotations: We thank the comment from Reviewer **RiBQ**, but **the annotation in the new proposed dataset remains at a high label accuracy**. Indeed, in the mentioned supplementary video, a total of 186 tracklets were showcased. After taking into account the two incorrect annotations that Reviewer RiBQ mentioned (ID 5274 and an ambiguous tracklet "man talking on a cell phone"), **the accurate annotation ratio remains at a high 98.9%** (184 out of 186).
> > Note that the annotations demonstrated in the submitted video are from version v0.6 (refer to **A13**). These specific instances have already been rectified in the new version v1.0 release.
> >
> >     - Appearance captions and action captions: To simplify visualization, we only showed either appearance or action captions type randomly as explained in Fig. 4's caption in the Supplementary PDF file.
> >
> >     - Inconsistent annotations in the same frame: These bikes are not included to be retrieved. We would like to emphasize that these variations, present an opportunity to examine the robustness and adaptability of the proposed type-to-track approach.
> >
> >     - Well-defined patterns: We appreciate your suggestion regarding the retrieval evaluation pattern. Given that the annotations represent **tracklet captions** that convey the essence of the tracklet, these captions remain accurate in terms of their visual meaning.
> >
> > - Underwhelming performance of MENDER: Please note that the proposed MENDER is **designed to address a new angle of visual-language MOT tracking problem** (as mentioned in L34-35 in the main paper) and is **NOT aimed to achieve the top rankings on the general MOT leaderboard**. The suggestion from the reviewer RiBQ is **NOT a fair comparison** since it is just to check the absolute bar. ByteTrack is a two-stage framework that utilizes an efficient single-category design in its separate object detector. In contrast, our design is a joint-detection-and-tracking framework agnostic to the category and for flexible textual input.
> >
> > - Lack of retrieval metrics: Type-to-track is **NOT solely a retrieval task**. As originally stated in the main paper L22-23 and L35-36, **the main and focused task is the new problem of prompt-based tracking**. The retrieval setting is an extension and the retrieval metric capturing the retrieval performance aspects is planned as our future iterations, as mentioned in the limitations section of our paper.

---

> > > ### Comment · Reviewer_9rR3 · 2023-08-14
> > > **Additional comment on the dataset quality**
> > >
> > > I also have a concern regarding the quality of captions in the dataset. As I mentioned in my review, is it valid to argue that this dataset contains "textual captions describing their appearance and action in detail" (L8)?
> > >
> > > In the annotations (the version available to the public now is v0.2), the captions are still not in order (appearance -> action) resulting in confusion about the type of captions whether it describes appearance or action. Also, the analysis of the diversity of captions is absent, for example, how many unique words are used for describing appearance or actions.
> > >
> > > Furthermore, in the MOT17 dataset, 20~30% of captions are sentences related to 'a person walking on the sidewalk' which is not informative compared to the original MOT annotations which already annotate the type of object, such as a walking pedestrian or standing person. I expect more fine-grained captions if they are annotated by professional annotators.

---

> > > ### Comment · Reviewer_RiBQ · 2023-08-14
> > >
> > > - Quality of annotations: I thank the authors for their reply, but I remain skeptical about the quality of the annotations. As reviewer 9rR3 pointed out, only v0.2 is available to the public as of 2023-08-14. We can only judge the dataset based on the current available version, and accepting a dataset paper with the promise that the dataset will be improved is a big bet.
> > >
> > > - Underwhelming performance of MENDER: The authors state that the comparison to ByteTrack is not fair. Even assuming that it isn't, MENDER loosely follows the design of MOTR. However, MOTR achieves considerably higher performance on MOT17, i.e. 65.0  vs. 74.5 MOTA  53.9 vs. 57.8 HOTA. What is this performance drop due to?
> > >
> > > - Lack of retrieval metrics: The authors themselves state that
> > > "[The variations in captions for the same type of object] present an opportunity to examine the robustness and adaptability of the proposed type-to-track approach.". However, I would argue that they may be helpful for training such models, but not "robustness and adaptability of the proposed type-to-track approach". The current class-agnostic metrics are not really helpful to evaluate the type-to-track task, and without a way to evaluate it, this dataset is useful only for training. The fact that the authors report the lack of retrieval metrics as a limitation in their paper does not discount from the fact that these metrics are necessary in this paper to evaluate the performance of the proposed task.

---

> > > > ### Author Response · Authors · 2023-08-15
> > > >
> > > > We sincerely appreciate the insightful feedback provided by the reviewers. Your inputs on various aspects, including the retrieval metrics and the quality of annotations, are invaluable. We are committed to incorporating these suggestions into the final version of our work to enhance its overall quality and impact.

---

> > > > > ### Author Response · Authors · 2023-08-17
> > > > > **Release of version v1.0 of the MOT17**
> > > > >
> > > > > We would like to extend our sincere gratitude for your invaluable efforts and insightful comments, which have helped improve our annotation quality. We are pleased to announce the release of version v1.0 of the MOT17 annotations. We welcome any further suggestions you may have, as we always strive to improve our annotations' quality.
> > > > >
> > > > > Here are the notable updates:
> > > > >
> > > > > - The annotation format adheres to our initial commitment. The `captions` field now includes the first caption for appearance and the second for action. Any missing captions have been filled with a `None` value.
> > > > >
> > > > > - Given that MOT17 videos dominantly feature crowded scenes with moving people, it is expected that most of the action captions pertain to walking and standing actions. We have endeavored to enhance diversity by introducing other actions like sitting, pulling, pushing, wandering, talking, etc. Furthermore, we have introduced specific actions like "using phone while walking" to enrich the range of captions. Please combine with the test annotations to comprehensively understand the diversity present.
> > > > >
> > > > > - The long-tailed distribution effect is less observed in appearance captions, adding to the overall balance and detail (L8).
> > > > >
> > > > > - Approximately 4.5% of total instances have no caption, while around 12.7% have only one caption associated with them due to the low visibility at a distance and instances of high occlusion.
> > > > >
> > > > > - The physical characteristics of a person or their personal accessories, such as their **clothing**, **bag color**, and **hair color** are considered to be part of their appearance. Therefore, the appearance captions include verbs carrying or holding to describe personal accessories.
> > > > >
> > > > > These changes will make the MOT17 annotations more comprehensive and informative, and we are grateful for your continued feedback.

---

> > > > > > ### Author Response · Authors · 2023-08-20
> > > > > > **Release training annotation of the TAO dataset**
> > > > > >
> > > > > > Dear reviewers,
> > > > > >
> > > > > > We deeply appreciate your constructive comments during the discussion. We are pleased to announce that ALL training annotations of version v1.0 have been released, including the MOT17 and TAO annotations.
> > > > > >
> > > > > > We welcome any further suggestions you may have, as we always strive to improve our annotations' quality.
> > > > > >
> > > > > > Here are the notable updates on TAO annotations:
> > > > > >
> > > > > > - Our annotation contains 948 unique words and 2591 unique sentences.
> > > > > >
> > > > > > - In terms of the well-defined pattern suggested by Reviewer RiBQ, we organize the captions as follows:
> > > > > >
> > > > > >     - Appearance caption of a highly visible object: a `[color]` `[subject]` in `[position]`
> > > > > >     - Appearance caption of a highly visible person: `["person"/"man"/"woman"]` `["wearing"/"carrying"/"in"]` `[color/pattern]` `[clothes]`.
> > > > > >     - Action caption of a highly visible person: `[subject]` `[verb]` in `[position/context]`.
> > > > > >
> > > > > > We sincerely appreciate your continuous engagement and dedication to improving our work.
> > > > > >
> > > > > > Regards,
> > > > > >
> > > > > > Authors

---

### Decision · Program_Chairs · 2023-09-21

**Decision:**

Accept (poster)

**Comment:**

This paper received accept, weak and borderline accept recommendations. There is a rebuttal.
Reviewer RiBQ has concerns around the dataset quality, which are shared with Reviewer 9rR3. The authors attempt to address these in the rebuttal. Reviewer 9rR3's concerns with dataset quality are addressed by the authors in the rebuttal and 9rR3 is willing to raise their score. Similarly, Reviewer h9ub's concerns around dataset quality have been sufficiently addressed in the final version of the dataset. Further h9ub thinks the performance of MENDER falls short of other SOTA MOT models but is sufficient for the purpose of this paper. Reviewer HDeT shares the dataset concerns and feels that the paper presentation is too complex; therefore remains their borderline accept rating. Reviewer j89C recommends borderline accept and the rebuttal resolves most of their concerns.
The AC agrees with the reviewers' opinion towards accepting this work. The authors are strongly encouraged to include the additional feedback into the paper and the improved dataset, as well as including the results from the rebuttal in the camera-ready version. Specifically, the authors should  conduct a comparison and discussion with RMOT in the revised version, including the performance comparison of MENDER to other SOTA trackers in the main paper.